# When More Experts Hurt: Underfitting in Multi-Expert Learning to Defer

Shuqi Liu [* 1]  Yuzhou Cao [* 1]  Lei Feng [2]  Bo An [1]  Luke Ong [1]

## Abstract

Learning to Defer (L2D) enables a classifier to abstain from predictions and defer to an expert, and has recently been extended to multi-expert settings. In this work, we show that multi-expert L2D is fundamentally more challenging than the single-expert case. With multiple experts, the classifier's underfitting becomes inherent, which seriously degrades prediction performance, whereas in the single-expert setting it arises only under specific conditions. We theoretically reveal that this stems from an intrinsic expert identifiability issue: learning which expert to trust from a diverse pool, a problem absent in the single-expert case and renders existing underfitting remedies failed. To tackle this issue, we propose PiCCE (**Pi**ck the **C**onfident and **C**orrect **E**xpert), a surrogate-based method that adaptively identifies a reliable expert based on empirical evidence. PiCCE effectively reduces multi-expert L2D to a single-expert–like learning problem, thereby resolving multi-expert underfitting. We further prove its statistical consistency and ability to recover class probabilities and expert accuracies. Extensive experiments across diverse settings, including real-world expert scenarios, validate our theoretical results and demonstrate improved performance.

## 1 Introduction

In risk-critical machine learning tasks, misclassification can be fatal. Unlike ordinary machine learning pipelines that deploy models solely for prediction, the Learning to Defer (**L2D**) paradigm (Madras et al., 2018; Mozannar & Sontag, 2020; Bansal et al., 2021) aims to enhance system reliability by integrating human experts' decisions. This framework

allows a system to defer the prediction of a sample to an expert if it deems the expert more capable of providing a correct prediction. Due to its practical importance, L2D has attracted significant attention in recent years, with substantial works extending the framework to more complex and realistic settings (De et al., 2021; Verma & Nalisnick, 2022; Charusaie et al., 2022; Verma et al., 2023; Mozannar et al., 2023; Mao et al., 2024b; Wei et al., 2024; Tailor et al., 2024; Charusaie & Samadi, 2024; Palomba et al., 2025; Montreuil et al., 2025a; Jitkrittum et al., 2023; 2025; Narasimhan et al., 2025).

Despite the conceptual appeal of L2D, training such systems remains computationally challenging. The performance is typically evaluated by the overall system accuracy, an objective that is non-convex and discrete, making it difficult to optimize directly. To render the training tractable, the dominant approach in the literature relies on continuous surrogate losses. These methods are often designed to satisfy statistical consistency, ensuring that optimizing the surrogate asymptotically recovers the optimal L2D rule (Mozannar & Sontag, 2020; Verma & Nalisnick, 2022; Cao et al., 2023; Liu et al., 2024; Ruggieri & Pugnana, 2025; Pugnana et al., 2025). Beyond surrogate-based training, alternative strategies such as post-hoc methods have also been proposed to bypass the direct optimization of the deferral policy (Narasimhan et al., 2022; Mao et al., 2023a; 2024a; Montreuil et al., 2025c;b).

While the majority of existing L2D research focuses on the single-expert setting, real-world applications often involve a pool of experts with complementary skills. This shift significantly increases the difficulty of the learning problem. In the single-expert case, the system only needs to make a binary decision of whether to predict or defer. In the multi-expert setting, however, the system must determine not only when to defer but also which specific expert is the most reliable for a given input. To address this challenge, Verma et al. (2023) extended classic single-expert methods (Mozannar & Sontag, 2020; Verma & Nalisnick, 2022) to the multi-expert domain. This approach was subsequently shown to be a special case of the unified consistency framework proposed by Mao et al. (2024a). In addition to these surrogate-based methods, Mao et al. (2023a; 2025) developed two-stage methods to explicitly handle expert selection processes.

*Equal contribution [1]College of Computing and Data Science, Nanyang Technological University, Singapore [2]School of Computer Science and Engineering, Southeast University, Nanjing, China. Correspondence to: Yuzhou Cao <yuzhou002@e.ntu.edu.sg>.

*Proceedings of the $43^{rd}$ International Conference on Machine Learning*, Seoul, South Korea. PMLR 306, 2026. Copyright 2026 by the author(s).

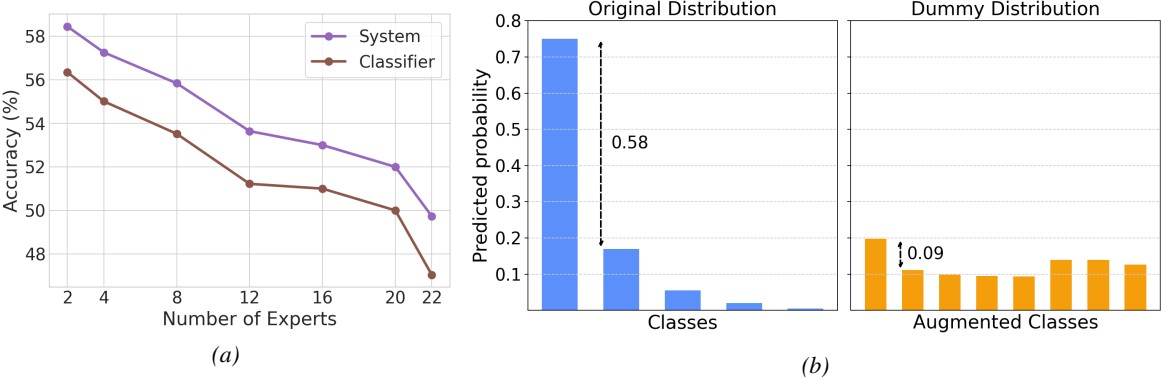

*Figure 1.* Left: Illustration of underfitting when using multi-expert CE surrogate loss proposed by Verma et al. (2023) on ImageNet. We consider a MobileNet-v2 model and progressively introduce "dog experts", where each expert covers a domain consisting of 5 dog species, attaining 85% accuracy on its domain, 75% on the other dog species, and random guessing on remaining classes. Since the experts have non-overlapping domains, adding more experts strictly increases the aggregate accuracy of the expert set. We report the test accuracy of both the system and the classifier. Right: An illustration of the degraded distribution for a 5-class classification task. We present the predicted class-posterior probabilities for an instance in descending order before and after the introduction of three experts.

While surrogate-based methods have proven effective in standard single-expert L2D settings, the phenomenon of classifier underfitting has been observed in a specific non-conventional scenario involving explicit deferral costs, where the weakened classifier significantly impairs the system's final accuracy and reliability. In this particular case, the issue arises from a redundant label-smoothing term induced by the cost parameter and has been successfully mitigated by specialized techniques (Narasimhan et al., 2022; Liu et al., 2024). Consequently, the standard cost-free setting is generally considered to be free from such issues.

Current research in the multi-expert L2D primarily focuses on selecting the optimal expert and typically operates under the conventional setting without extra deferral costs. However, we identify that classifier underfitting unexpectedly persists in this general multi-expert setting (as in Figure 1a) despite the complete absence of deferral costs. Our analysis attributes this to the *expert aggregation term* inherent to multi-expert objectives rather than the cost-induced smoothing in single-expert cases. Since the underlying cause is fundamentally different, previous remedies are thus inapplicable, highlighting the need for alternative solutions.

To address these issues, we propose the **Picking the Confident and Correct Expert** (PiCCE) loss formulation, which mitigates the underfitting issues caused by the expert aggregation term, by exploiting the empirical/ground-truth information. We further provide theoretical guarantees for the proposed PiCCE from both optimization and statistical efficiency perspectives. Our main contributions are:

- In Section 3, we identify that classifier underfitting persists in the general multi-expert setting even without deferral costs, which contradicts the intuition from single-expert L2D. We theoretically attribute this phe-

nomenon to the *expert aggregation term*, which fundamentally differs from cost-induced label smoothing observed in prior literature.

- Given that the distinct cause of underfitting renders prior remedies ineffective, we introduce *PiCCE* in Section 4, a surrogate-based method that dynamically selects experts in a data-dependent manner. We demonstrate that it enjoys favorable optimization properties and resolves underfitting by significantly compressing the expert aggregation term.

- In Section 5, we analyze the statistical consistency of PiCCE. We prove that PiCCE is not only consistent but also capable of accurately recovering both the underlying class probabilities and expert accuracies.

- We experimentally verify the underfitting caused by expert aggregation in Section 6 and demonstrate the effectiveness of PiCCE using both synthetic and real-world experts.

## 2  Preliminaries

In this section, we first introduce the problem formulation of L2D with multiple experts and existing solutions, then review the issue of underfitting under single-expert setting.

### 2.1  Problem Formulation and Existing Losses for L2D with Multiple Experts

**Data Generating Distribution:** Denote by $\mathcal{X} \subseteq \mathbb{R}^d$ the feature space, $\mathcal{Y} := [K]$ the label space, and $\mathcal{M} = \mathcal{Y}$ the expert prediction space. The expert prediction space for $J$ experts is $\mathcal{M}^J = [K]^J$, and $m_j$ is the prediction of the $j$-th expert for any $\boldsymbol{m} \in \mathcal{M}^J$. The data generating distribution

of L2D to multiple experts is $\mathcal{D}$ on $\mathcal{X} \times \mathcal{Y} \times \mathcal{M}^J$, and the density of this distribution is $p(\boldsymbol{x}, y, \boldsymbol{m})$. We assume the training data set consists of $n$ i.i.d. samples drawn from $\mathcal{D}$.

**Technical Notations:** Let us write $\eta_y(\boldsymbol{x}) := \Pr(Y = y \mid X = \boldsymbol{x})$. The $i$-th expert's prediction accuracy at point $\boldsymbol{x}$ is denoted by $\text{Acc}_i(\boldsymbol{x}) := \Pr(M_i = Y \mid X = \boldsymbol{x})$. We define the $\text{argmax}_{i \in \mathcal{I}} \theta_i \in \mathcal{I}$ that breaks ties arbitrarily and deterministically, and $\text{Argmax}$ is the original set of maximum indices. Define the augmented label space as $\mathcal{Y}^\perp = \mathcal{Y} \cup \{\perp_1, \cdots, \perp_J\}$, where $\perp_j$ refers to the option of deferring to the $j$-th expert.

**Problem Setup:** The aim of L2D with multiple experts is to learn a model $f : \mathcal{X} \to \mathcal{Y}^\perp$ with performance evaluated by the following *target system loss* (Verma et al., 2023):

$$\ell_{01}^\perp(f(\boldsymbol{x}), y, \boldsymbol{m}) := \begin{cases} \mathbb{1}[f(\boldsymbol{x}) \neq y], & \text{if } f(\boldsymbol{x}) \in \mathcal{Y}, \\ \mathbb{1}[m_j \neq y], & \text{if } f(\boldsymbol{x}) = \perp_j, \end{cases} \quad (1)$$

where $\mathbb{1}[f(\boldsymbol{x}) \neq y]$ and $\mathbb{1}[m_j \neq y]$ represent the zero-one losses of the classifier and the $j$-th expert, respectively, taking value of 1 if the corresponding prediction is incorrect. The *risk minimization problem* w.r.t. (1) is defined as below,

$$\min_f R_{01}^\perp(f) = \mathbb{E}_{p(\boldsymbol{x}, y, \boldsymbol{m})}[\ell_{01}^\perp(f(\boldsymbol{x}), y, \boldsymbol{m})], \quad (2)$$

where the target risk $R_{01}^\perp(f)$ is the misclassification error of the whole L2D system over $\mathcal{D}$. Its *Bayes optimal solution* is

$$f^*(\boldsymbol{x}) = \begin{cases} \perp_{j^*}, & \text{if } \text{Acc}_{j^*}(\boldsymbol{x}) \geq \eta_{y^*}(\boldsymbol{x}), \\ \text{argmax}_{y \in [K]} \eta_y(\boldsymbol{x}), & \text{else}, \end{cases} \quad (3)$$

where $y^* := \text{argmax}_{y \in [K]} \eta_y(\boldsymbol{x})$ denotes the optimal label, and $j^* := \text{argmax}_{j \in [J]} \text{Acc}_j(\boldsymbol{x})$ is the most accurate expert. Intuitively, the optimal L2D system triggers deferral options when an expert outperforms the classifier, ensuring each decision is handled by the most competent entity.

**Existing Loss Frameworks:** Given the discontinuous nature of the multi-expert risk (2), which is similar to the single-expert case as the multi-expert setting follows the same development, surrogate losses have been proposed to make the optimization problem tractable. While Hemmer et al. (2023) introduced a mixture of experts method, Verma et al. (2023) showed that it is inconsistent and overcame this limitation by generalizing single-expert CE (Mozannar & Sontag, 2020) and OvA (Verma & Nalisnick, 2022) surrogate losses to the multiple-expert setting, which can be further unified into a general framework (Mao et al., 2023a):

$$\ell_\phi(\boldsymbol{\theta}, y, \boldsymbol{m}) := \phi(\boldsymbol{\theta}, y) + \sum_{j=1}^J \mathbb{1}[m_j = y]\, \phi(\boldsymbol{\theta}, j + K) \quad (4)$$

where $\phi : \mathbb{R}^{K+J} \times [K+J] \to \mathbb{R}_{\geq 0}$ is a multiclass loss function and $\boldsymbol{\theta} \in \mathbb{R}^{K+J}$ denotes the *score vector* produced by

the model for input $\boldsymbol{x}$ via a scoring function $g : \mathcal{X} \to \mathbb{R}^{K+J}$, i.e., $g(\boldsymbol{x}) = \boldsymbol{\theta}$. For brevity, we omit the dependence of $\boldsymbol{\theta}$ on $\boldsymbol{x}$ in the rest of this paper, w.l.o.g., since the risk is defined pointwise as an expectation over $\boldsymbol{x}$. Note that (4) is consistent provided that the multi-class surrogate loss $\phi$ is consistent (Charusaie et al., 2022; Mao et al., 2024a), i.e., minimizing the expected surrogate risk w.r.t. (4) recovers the Bayes optimal solution (3). The L2D system $f$ is implemented via the composition $\varphi \circ g$, i.e., $f(\boldsymbol{x}) = (\varphi \circ g)(\boldsymbol{x})$, where $\varphi : \mathbb{R}^{K+J} \to \mathcal{Y}^\perp$ is a prediction link defined as:

$$\varphi(\boldsymbol{\theta}) = \begin{cases} \text{argmax}_y \theta_y, & \text{if } \text{argmax}_y \theta_y \in [K], \\ \perp_{(\text{argmax}_y \theta_y - K)}, & \text{else}. \end{cases} \quad (5)$$

### 2.2 Underfitting Issues in L2D

When $J = 1$, the above problem reduces to the single-expert L2D setting (Mozannar & Sontag, 2020; Verma & Nalisnick, 2022; Cao et al., 2023). A special studied case further incorporates a non-zero deferral cost $c$ (Narasimhan et al., 2022; Liu et al., 2024), reflecting the practical constraints that expert consultation often incurs additional computational or monetary expenses. Under this formulation, compared with (1), the expert-related loss becomes $\mathbb{1}[m \neq y] + c$. Consequently, the optimal system defers only when the expert's accuracy exceeds the classifier's by at least a margin of $c$.

However, the inclusion of $c > 0$ has been found to trigger classifier underfitting (Narasimhan et al., 2022; Liu et al., 2024). This issue is largely rooted in the learning objective's structure: specifically, Narasimhan et al. (2022) identified that in both CE (Mozannar & Sontag, 2020) and OvA-based (Verma & Nalisnick, 2022) surrogates, the presence of $c$ introduces a (redundant) label-smoothing term (Szegedy et al., 2016) that hampers the classifier's learning. Furthermore, Liu et al. (2024) showed that this cost tends to induce a flattened label distribution that scales with the number of classes $K$, making the classifier more likely to incorrectly swap the optimal label with a competing one.

To mitigate these $c$-induced underfitting issues, a post-hoc estimator was proposed by Narasimhan et al. (2022). Inspired by the theoretical success of the end-to-end approaches (Mozannar & Sontag, 2020; Verma & Nalisnick, 2022), Liu et al. (2024) further provided a one-staged loss framework to eliminate the label-redundant term by utilizing the intermediate learning results.

## 3 More Experts, Worse Performance: A New Underfitting Challenge

In Section 2.2, we reviewed the underfitting issues under the single-expert with non-zero deferral costs setting. We now show that this underfitting problem persists in the multi-expert scenario even without additional deferral costs, and

show that existing methods and seemingly plausible extensions based on them failed in this case.

## 3.1 Multi-Expert Can Cause Underfitting

We focus on the multi-expert L2D framework (4) in the standard setting without additional deferral costs. Under Bayes optimality (3), the system always selects the most accurate predictor from the pool of experts and the classifier. Consequently, a larger expert set should never lead to a decrease in optimal performance. For instance, consider two expert sets $\mathcal{E}_1 \subset \mathcal{E}_2$, the best-in-set accuracy satisfies:

$$\max_{j \in \mathcal{E}_2} \mathrm{Acc}_j(\boldsymbol{x}) \geq \max_{j \in \mathcal{E}_1} \mathrm{Acc}_j(\boldsymbol{x}),$$

which implies that the optimal L2D system induced from $\mathcal{E}_2$ will be better than or equal to the one from $\mathcal{E}_1$.

However, the empirical evidence in Figure 1a reveals an unexpected inversion of this analysis: expanding the expert set induces *underfitting* in the base classifier, which subsequently leads to a decline in the overall system accuracy. This indicates that a larger and more capable expert set can actually compromise the learning process, causing the system to underperform as more experts are introduced.

This finding is particularly notable because (4) is free from redundant label-smoothing and should have avoided underfitting (Narasimhan et al., 2022; Liu et al., 2024). We thus identify a novel multi-expert underfitting challenge that contradicts these existing conclusions.

How can a more capable expert set induce such underfitting? To uncover the underlying mechanism, we begin by analyzing the risk w.r.t. (4).

$$R_{\ell_\phi|\boldsymbol{x}}(\boldsymbol{\theta}) = \sum_{y=1}^{K} \eta_y(\boldsymbol{x})\phi(\boldsymbol{\theta}, y) + \sum_{j=1}^{J} \mathrm{Acc}_j(\boldsymbol{x})\phi(\boldsymbol{\theta}, j+K).$$

The above risk formulation is closely related to a $(K+J)$-class classification problem over dummy distribution $\widehat{p}$, where the augmented class probabilities for a given $\boldsymbol{x}$ is $\widehat{\boldsymbol{\eta}}(\boldsymbol{x}) = [\widehat{p}(1\,|\,\boldsymbol{x}), \cdots, \widehat{p}(K\,|\,\boldsymbol{x}), \widehat{p}(\perp_1|\,\boldsymbol{x}), \cdots, \widehat{p}(\perp_J|\,\boldsymbol{x})]$, and

$$\widehat{p}(y|\boldsymbol{x}) = \frac{\eta_y(\boldsymbol{x})}{1 + \mathcal{A}(\boldsymbol{x})}, \quad \widehat{p}(\perp_j|\boldsymbol{x}) = \frac{\mathrm{Acc}_j(\boldsymbol{x})}{1 + \mathcal{A}(\boldsymbol{x})}, \quad (6)$$

where *the expert aggregation term* $\mathcal{A}(\boldsymbol{x}) = \sum_{j=1}^{J} \mathrm{Acc}_j(\boldsymbol{x})$ captures the sum of experts' accuracy. Observe that the expert aggregation term $\mathcal{A}(\boldsymbol{x})$ scales as $\mathcal{O}(J)$ with the number of experts. Unlike the single-expert case where $\mathcal{A}(\boldsymbol{x}) = \mathcal{O}(1)$, this multi-expert term increasingly flattens the label distribution $\widehat{p}(y|\boldsymbol{x})$. As Figure 1b illustrates, this flattening narrows the margin between the optimal label and its competitors, making the ground truth harder to identify and eventually triggering underfitting.

In short, flattened label distribution arises inevitably in multi-expert settings, which is quite different from single-expert cases. Consequently, this inherent flattening reveals that existing losses like CE and OvA (Verma et al., 2023) will suffer from underfitting due to their intrinsic objective structure rather than external assumptions, e.g., deferral costs.

## 3.2 Failure of Merely Using Intermediate Results

As established in Section 3.1, underfitting in multi-expert L2D stems from the expert aggregation term rather than redundant label smoothing. Consequently, existing underfitting-resistant approaches (Liu et al., 2024) designed to eliminate label smoothing fail to resolve the underfitting inherent to the multi-expert setting, which necessitates new methods tailored for the multi-expert underfitting.

Since the aggregation term $\mathcal{A}(\boldsymbol{x}) = \mathcal{O}(J)$ is the primary cause of underfitting, a direct solution is to reduce the multi-expert system to a single-expert setup. To test this logic, consider an idealized scenario where the most accurate expert $j^*$ for an instance is known. In this case, comparing the classifier solely against $j^*$ substitutes $\mathcal{A}(\boldsymbol{x})$ with $\mathrm{Acc}_{j^*}(\boldsymbol{x})$, which preserves Bayes optimality while immediately eliminating the label-flattening effect. Although $j^*$ is inaccessible in practice, existing work (Liu et al., 2024) suggests that intermediate learning results can serve as a proxy of the optimal expert, i.e., utilizing model's predictions at intermediate stages of the training process instead of the true optimal expert $j^*$. This motivates the Pick the Confident Expert method below following the logic of (Liu et al., 2024):

$$\widetilde{\ell}_\phi^\circ(\boldsymbol{\theta}, y, \boldsymbol{m}) := \phi(\boldsymbol{\theta}, y) + \mathbb{1}[m_{\widehat{j}^*} = y]\phi(\boldsymbol{\theta}, \widehat{j}^* + K), \quad (7)$$

where $\widehat{j}^* = \mathrm{argmax}_{j \in [J]}\, \theta_{j+K}$ is an estimator of the optimal expert, and the corresponding conditional risk is:

$$R_{\widetilde{\ell}_\phi^\circ|\boldsymbol{x}}(\boldsymbol{\theta}) = \sum_{y=1}^{K} \eta_y(\boldsymbol{x})\phi(\boldsymbol{\theta}, y) + \mathrm{Acc}_{\widehat{j}^*}(\boldsymbol{x})\phi(\boldsymbol{\theta}, \widehat{j}^* + K).$$

By reducing the expert aggregation in (4) to a single estimated optimal expert term, this new formulation (7) induces a less flattened (dummy) label distribution $p'$:

$$p'(y|\boldsymbol{x}) = \frac{\eta_y(\boldsymbol{x})}{1 + \mathrm{Acc}_{\widehat{j}^*}(\boldsymbol{x})}, p'(\perp_j|\boldsymbol{x}) = \frac{\mathrm{Acc}_j(\boldsymbol{x})}{1 + \mathrm{Acc}_{\widehat{j}^*}(\boldsymbol{x})}. \quad (8)$$

However, despite its success in mitigating label flattening, this seemingly reasonable solution *fails* from an optimization perspective because it lacks *continuity*, a fundamental property of any viable surrogate loss. Specifically, the indicator term in (7) is non-constant and depends on the scores $\boldsymbol{\theta}$ through the discrete estimator $\widehat{j}^*$. Consequently, any shift in the selected expert $\widehat{j}^*$ induces a step discontinuity, thereby precluding effective gradient-based optimization.

# 4 PiCCE: Using Both Intermediate and Empirical Results

According to the discussion in Section 3, the multi-expert L2D paradigm is naturally prone to underfitting compared with single-expert L2D, while existing surrogate-based strategy for mitigating underfitting fails to achieve continuity, which violates the essential premise of surrogate loss design.

In this section, we show that this new challenge can be effectively solved by *exploiting the empirical/ground-truth information,* i.e., considering both the confidence and the *empirical correctness* of the experts, in the surrogate loss design. The further integration of ground-truth information not only enables continuous surrogates, but also greatly mitigates the unique phenomenon of underfitting caused by expert aggregation in multi-expert L2D.

## 4.1 Regulating Confident Experts with Ground-truth

To achieve a continuous surrogate framework, let's analyze the reason for the discontinuity of (7) first. Revisiting the definition of expert selector $\widehat{j}^*$ in (7), it is noticeable that the selection ranges over the complete set of experts $[J]$, which encompasses both accurate and erroneous predictions. When the most confident expert shifts among this full candidate set, (7) may encounter step-like discontinuities due to the indicator function $\mathbb{1}[m_{\widehat{j}^*} = y]$, if the correctness of selected experts varies after the shift.

This observation raises a concern: is it really helpful to consider the whole expert set? According to the discussion above, choosing the full set $[J]$ indiscriminately contributes to the step-like discontinuities inherent in (7), which suggests the potential benefit of **constraining the expert set**. Furthermore, it is also intuitive to prune the expert set based on **empirical observations** of their performance, thereby focusing the selections on high-accuracy experts.

Therefore, a natural idea is to modify the expert selector based on the intuition of regulating the candidate set with empirical evidences of expert accuracy. Despite potential concerns regarding the cost of acquiring such evidence, the ground-truth $y$ and expert predictions $m_j$ serve as surprisingly straightforward sources. Specifically, they constitute the unbiased estimator $\mathbb{1}[m_j = y]$ of expert accuracy $\text{Acc}_j$. Moreover, this incurs zero additional cost, given that they are readily available within the standard L2D training set.

Motivated by this empirical correctness evidence $\mathbb{1}[m_j = y]$, we proceed to construct a more compact expert selector: confining the expert set to the correct experts $\{j \in [J] : m_j = y\}$, and selecting the most confident expert in this set. Compared with the pick the confident expert method (7) that chooses from the full set $[J]$, our proposed new selector operates in a manner of *Picking the Confident and Correct*

*Expert* (PiCCE), which induces the following surrogates:

**Definition 4.1** (PiCCE). Denote by $[\boldsymbol{m} = y]$ the set $\{j \in [J] : m_j = y\}$. For any multiclass loss $\phi : \mathbb{R}^{K+J} \times [K + J] \to \mathbb{R}_{\geq 0}$, our loss $\widetilde{\ell}_\phi : \mathbb{R}^{K+J} \times \mathcal{Y} \times \mathcal{M}^J \to \mathbb{R}_{\geq 0}$ is:

$$\widetilde{\ell}_\phi(\boldsymbol{\theta}, y, \boldsymbol{m}) = \phi(\boldsymbol{\theta}, y) + \phi\Big(\boldsymbol{\theta}, \underset{j \in [\boldsymbol{m}=y]}{\operatorname{argmax}} \, \theta_{j+K} + K\Big). \quad (9)$$

The second term is 0 if $[\boldsymbol{m} = y] = \emptyset$.

The most intuitive difference between PiCCE (9) and the discontinuous loss (7) is that (9) **removes the multiplicative indicator term.** However, this is not the result of manual exclusion, but a natural consequence of the transition of candidate expert sets: for any expert $j \in [\boldsymbol{m} = y]$, the indicator term $\mathbb{1}[m_j = y]$ equals 1, and thus naturally integrates into the formulation (9). The benefit of this distinction is obvious since it removes the potential step-like discontinuities, which enables the construction of continuous surrogates:

**Theorem 4.2** (Continuity of PiCCE). *The proposed formulation (9) is continuous if $\phi$ is continuous and is symmetric w.r.t. its last $J$ inputs, i.e., $P\phi(\boldsymbol{\theta}) = \phi(P\boldsymbol{\theta})$ for permutation matrices $P \in \mathbb{R}^{K+J \times K+J}$ that $P_{i,i} = 1$ for $i \in [K]$, and $\phi(\boldsymbol{\theta}) = [\phi(\boldsymbol{\theta}, 1), \cdots, \phi(\boldsymbol{\theta}, K+J)]^\top$.*

The symmetry requirement on the base multiclass loss $\phi$ is mild, and it covers most multiclass losses used in existing multi-expert L2D losses (Verma et al., 2023; Mao et al., 2024a). For example, cross-entropy loss is a natural choice that satisfies the symmetry, which is used in the multi-expert CE loss (Verma et al., 2023; Mozannar & Sontag, 2020). Similarly, the base loss for the multi-expert OvA loss, as we will show in Section 5, also satisfies the symmetry.

## 4.2 Underfitting-resistance of PiCCE

While we have shown that the refinement of candidate expert set yields continuity, which is beneficial from an optimization perspective, a more critical issue lies in its statistical efficiency, in particular its robustness against underfitting, which constitutes the primary target of our design.

In this section, we show that PiCCE is indeed *underfitting-resistant* by demonstrating its dummy distribution. To begin with, we first analyze the risk formulation of PiCCE:

**Lemma 4.3** (Risk of PiCCE). *Suppose $\phi$ is symmetric w.r.t. its last $J$ inputs. Denote by $\boldsymbol{\sigma}$ any permutation of $[J]$ such that $\theta_{\sigma_1+K} \geq \cdots \geq \theta_{\sigma_J+K}$. Then the risk of PiCCE is:*

$$R_{\widetilde{\ell}_\phi|\boldsymbol{x}}(\boldsymbol{\theta}) = \sum_{y=1}^{K} \eta_y(\boldsymbol{x})\phi(\boldsymbol{\theta}, y) + \sum_{j=1}^{J} A_{\boldsymbol{\sigma}}^j(\boldsymbol{x})\phi(\boldsymbol{\theta}, \sigma_j + K) \quad (10)$$

*where $A_{\boldsymbol{\sigma}}^j(\boldsymbol{x}) = \Pr(M_{\sigma_{1:j-1}} \neq Y, M_{\sigma_j} = Y | X = x)$.*

This risk formulation is also a weighted sum of multiclass losses, which allows an analysis based on its corresponding

$(K + J)$-class dummy distribution $\widetilde{p}$. Specifically, its label counterpart ($y \in [K]$) can be formulated as:

$$\widetilde{p}(y|\boldsymbol{x}) = \frac{\eta_y(\boldsymbol{x})}{\sum\limits_{y=1}^{K} \eta_y(\boldsymbol{x}) + \sum\limits_{j=1}^{J} A_{\boldsymbol{\sigma}}^j(\boldsymbol{x})} = \frac{\eta_y(\boldsymbol{x})}{1 + \sum\limits_{j=1}^{J} A_{\boldsymbol{\sigma}}^j(\boldsymbol{x})}$$

While the dummy distribution can be explicitly formulated, the denominator $1 + \sum_{j=1}^{J} A_{\boldsymbol{\sigma}}^j(\boldsymbol{x})$ is less intuitive than those in (6) and (8). Specifically, the aggregation term $\mathcal{A}(\boldsymbol{x})$ in (6) clearly scales as $\mathcal{O}(J)$ because it represents the sum of expert accuracies, while the corresponding term $\mathrm{Acc}_{\widehat{j}*}(\boldsymbol{x})$ in (8) is simply $\mathcal{O}(1)$. In contrast, the term $\sum_{j=1}^{J} A_{\boldsymbol{\sigma}}^j(\boldsymbol{x})$ is significantly harder to quantify. While it superficially appears to be $\mathcal{O}(J)$ as a summation of $J$ terms, the internal dependencies between the $A_{\boldsymbol{\sigma}}^j(\boldsymbol{x})$ components suggest the potential for further simplification. This lack of transparency complicates the study of underfitting, particularly when quantifying the degree of label distribution flattening.

Fortunately, the following lemma indicates $\sum_{j=1}^{J} A_{\boldsymbol{\sigma}}^j(\boldsymbol{x})$ yields a clear formulation by exploiting their dependencies:

**Lemma 4.4.** *For any permutation $\boldsymbol{\sigma}$ and $\boldsymbol{x} \in \mathcal{X}$:*

$$1 + \sum_{j=1}^{J} A_{\boldsymbol{\sigma}}^j(\boldsymbol{x}) = 1 + \Pr\Big( \bigcup_{j \in [J]} M_j = Y \mid X = \boldsymbol{x} \Big). \quad (11)$$

Compared with dummy distribution (6) that severely flattens the label distribution since $\mathcal{A}(\boldsymbol{x})$ can increase up to $J$, Lemma 4.4 indicates that PiCCE successfully resolves this issue by compressing the sum of expert accuracy $\mathcal{A}(\boldsymbol{x}) = \mathcal{O}(J)$ into $\Pr\Big( \bigcup_{j \in [J]} M_j = Y \mid X = \boldsymbol{x} \Big) = \mathcal{O}(1)$, which leads to a dummy distribution that remains informative with increasing expert number, and thus is free from the multi-expert underfitting issue. In the next section, we further analyze the statistical consistency of PiCCE, i.e., if the minimization of the risk (10) can recover the Bayes optimal solution for multi-expert L2D (3).

## 5 Consistency Guarantee

In Section 5.1, we first show that the classifier counterpart in the L2D system is guaranteed to be consistent, i.e., the label with the highest score is always the most probable label. Then we further justify the consistency of the whole L2D system under a mild condition. We also analyze the finite-sample guarantee of PiCCE in Appendix E. Besides, our analyses focus on the following two multiclass losses:

$$\phi(\boldsymbol{\theta}, y) = -\log \mathrm{softmax}(\boldsymbol{\theta})_y, \quad (12)$$

where $\mathrm{softmax}(\boldsymbol{\theta})_y = \frac{\exp(\theta_y)}{\sum_{y=1}^{K+J} \exp(\theta_y)}$,

$$\phi(\boldsymbol{\theta}, y) = \begin{cases} -\log s(\theta_y) + \log(1 - s(\theta_y)), \, y > K, \\ -\log s(\theta_y) - \sum\limits_{y' \neq y} \log(1 - s(\theta_{y'})), \, \text{else}, \end{cases} \quad (13)$$

where $s(x) = \frac{1}{1 + \exp(-x)}$ is the sigmoid function. When (12) and (13) is used in (4), they recover the multi-expert CE/OvA-log losses (Verma et al., 2023), respectively.

### 5.1 Consistency Analysis of the Classifier

We analyze the properties of the first $K$ dimensions of the optimal scoring function, i.e., the optimal classifier, which is characterized by the following lemma:

**Lemma 5.1** (Consistency of Optimal Classifiers). *When using (12) and (13) in (9), their corresponding risk (10) is minimizable for any $\boldsymbol{x} \in \mathcal{X}$. Furthermore, denote by $\boldsymbol{\theta}^*$ any minimizer of (10) and any $\boldsymbol{x} \in \mathcal{X}$, when (12) and (13) is used as multiclass loss $\phi$ in PiCCE (9), $\arg\max_{y \in [K]} \theta_y^* \in \mathrm{Argmax}_{y \in [K]} \eta_y(\boldsymbol{x})$, and:*

*(A). When (12) is used in (9), denote by $\eta_y^* = \mathrm{softmax}(\boldsymbol{\theta}^*)_y$ and $u_j^* = \mathrm{softmax}(\boldsymbol{\theta}^*)_{K+j}$:*

$$\eta_y^* = \eta_y(\boldsymbol{x})(1 - \widetilde{V}(\boldsymbol{x})), \quad \forall y \in [K].$$

*where $\widetilde{V}(\boldsymbol{x}) := \sum_{j=1}^{J} u_j^* = \frac{V(\boldsymbol{x})}{1 + V(\boldsymbol{x})}$ and $V(\boldsymbol{x}) = \Pr(\bigcup_{j \in [J]} M_j = Y | X = \boldsymbol{x})$.*

*(B). When (13) is used in (9), $s(\theta_y^*) = \eta_y(\boldsymbol{x}), \forall y \in [K]$.*

This lemma immediately establishes the consistency of PiCCE with respect to (12) and (13), implying that the optimal L2D system serves as an effective classifier. Furthermore, the method yields Fisher-consistent class probability estimators. Specifically, Lemma 5.1 (A) shows that when (12) is used for PiCCE, the term $\frac{\mathrm{softmax}(\boldsymbol{\theta})_y}{1 - \sum_{i=K+1}^{K+J} \mathrm{softmax}(\boldsymbol{\theta})_i}$ converges to the class probability $\eta_y(\boldsymbol{x})$ as $\boldsymbol{\theta}$ approaches $\boldsymbol{\theta}^*$. Analogously, Lemma 5.1 (B) demonstrates that $s(\theta_y)$ converges to $\eta_y(\boldsymbol{x})$ when the OvA-log loss (13) is employed.

### 5.2 Consistency Analysis of the L2D System

Based on the consistency of the classifier in Section 5.1, we now examine the consistency of the whole L2D system, i.e., whether the system can recognize the best expert and choose the better one between it and the optimal classifier.

For any $\mathcal{M}' \subseteq \mathcal{M}$, $C_{\mathcal{M}'} := \{\exists j \in \mathcal{M}' : M_j = Y\}$ is the event that there exists correct experts in $\mathcal{M}'$. We show that consistency holds based on the following mild condition:

**Condition 1** (Information Advantage of Optimal Expert). *For any $\boldsymbol{x} \in \mathcal{X}$, assume the optimal expert is unique, i.e.,*

*Table 1.* The mean of the system error (Err, rescaled to 0-100) and coverage (Cov) for 3 trails on the CIFAR-100 and ImageNet datasets.

| Dataset | | CIFAR-100 | | | | | | | ImageNet | | | | | | | |
|---|---|---|---|---|---|---|---|---|---|---|---|---|---|---|---|---|
| Expert Pattern | | Animal Expert | | | | Overlapped Animal Expert | | | | Dog Expert | | | | Overlapped Dog Expert | | |
| Loss Formulation | | Vanilla | | PiCCE | | Vanilla | | PiCCE | | Vanilla | | PiCCE | | Vanilla | | PiCCE | |
| Method | #Exp | Err | Cov | Err | Cov | Err | Cov | Err | Cov | Err | Cov | Err | Cov | Err | Cov | Err | Cov |
| CE | 4 | 18.48 | 74.58 | **18.32** | **77.50** | 16.10 | 64.50 | **15.10** | **67.80** | 42.74 | 89.08 | **42.10** | **89.85** | 41.56 | 88.37 | **41.23** | **89.16** |
|  | 8 | 18.91 | 74.35 | **18.21** | **76.35** | 16.24 | 64.59 | **16.10** | **71.35** | 44.16 | 88.71 | **41.96** | **89.91** | 42.65 | 87.99 | **41.72** | **89.23** |
|  | 12 | 19.08 | 75.18 | **18.19** | **77.43** | 16.99 | 66.09 | **15.84** | **69.86** | 46.36 | 88.50 | **41.50** | **89.78** | 44.37 | 87.85 | **41.77** | **89.14** |
|  | 16 | 19.14 | 76.39 | **18.16** | **79.14** | 18.72 | 67.21 | **15.61** | **73.62** | 47.01 | 88.45 | **41.40** | **89.99** | 45.17 | 87.54 | **41.79** | **88.94** |
|  | 20 | 21.13 | 73.57 | **18.11** | **78.33** | 19.22 | 69.31 | **15.58** | **73.95** | 48.00 | 88.51 | **41.32** | **90.01** | 46.37 | 87.34 | **42.07** | **88.80** |
| OvA | 4 | 19.46 | 83.72 | **19.10** | **86.43** | 17.07 | 79.28 | **16.58** | **83.85** | 45.77 | 88.45 | **42.41** | **89.17** | 44.56 | 87.30 | **42.03** | **88.30** |
|  | 8 | 20.09 | 83.30 | **18.98** | **86.83** | 18.03 | 79.69 | **17.71** | **83.45** | 52.62 | 86.66 | **42.11** | **89.29** | 52.05 | 85.56 | **42.15** | **88.51** |
|  | 12 | 20.70 | 82.67 | **18.86** | **86.89** | 18.45 | 79.36 | **17.77** | **84.93** | 57.66 | 85.74 | **42.45** | **89.20** | 56.64 | 84.51 | **42.20** | **88.74** |
|  | 16 | 21.84 | 83.09 | **18.70** | **87.11** | 19.85 | 77.30 | **17.76** | **85.02** | 62.16 | 84.67 | **42.17** | **89.12** | 59.43 | 83.68 | **42.03** | **88.70** |
|  | 20 | 22.91 | 77.71 | **18.63** | **86.69** | 20.95 | 72.72 | **17.74** | **86.19** | 66.56 | 83.90 | **42.25** | **89.08** | 62.91 | 82.93 | **42.10** | **88.77** |

*Table 2.* The mean of the system error (Err, rescaled to 0-100) and coverage (Cov) for 3 trails on the MiceBone and Chaoyang datasets.

| Dataset | MiceBone | | | | | | | | Chaoyang | | | | | | | |
|---|---|---|---|---|---|---|---|---|---|---|---|---|---|---|---|---|
| Method | CE | | PiCCE-CE | | OvA | | PiCCE-OvA | | CE | | PiCCE-CE | | OvA | | PiCCE-OvA | |
| #Exp | Err | Cov | Err | Cov | Err | Cov | Err | Cov | Err | Cov | Err | Cov | Err | Cov | Err | Cov |
| 2 | 15.17 | 60.92 | **15.23** | **69.28** | 14.91 | 72.26 | **14.06** | **83.34** | 1.32 | 53.86 | **1.22** | **63.05** | 1.51 | 32.33 | **1.46** | **37.76** |
| 4 | 15.88 | 55.09 | **15.17** | **68.11** | 15.02 | 68.24 | **13.80** | **70.33** | 1.75 | 51.68 | **1.60** | **55.86** | 1.90 | 20.37 | **1.56** | **27.76** |
| 6 | 15.99 | 51.11 | **14.97** | **62.22** | 15.89 | 63.84 | **13.09** | **66.10** | 2.48 | 45.99 | **1.75** | **52.02** | 2.04 | 14.39 | **1.31** | **23.58** |
| 8 | 16.72 | 43.62 | **13.35** | **60.72** | 16.72 | 57.62 | **13.03** | **59.96** | 3.35 | 41.23 | **2.04** | **50.90** | 2.48 | 13.17 | **1.02** | **20.38** |

$|\mathrm{Argmax}_{j\in[J]}\mathrm{Acc}_j(\boldsymbol{x})| = 1$ *and we denote it as $j^*$. For any expert $j \neq j^*$ and any expert set $\mathcal{M}' \subseteq [J]/\{j, j^*\}$:*

$$\Pr\left(C_{\mathcal{M}'\cup\{j^*\}}|X = \boldsymbol{x}\right) > \Pr\left(C_{\mathcal{M}'\cup\{j\}}|X = \boldsymbol{x}\right) \quad (14)$$

In essence, this condition implies that the optimal expert yields the greatest improvement to the system's potential accuracy. This is a natural expectation in practical scenarios, as the optimal expert typically possesses predictive advantages that are not fully subsumed by the collective knowledge of the other experts. A fundamental example is the deterministic expert setting (Mao et al., 2023a; 2024a), where each expert outputs a fixed, distinct label. In this case, incorporating the optimal expert (who predicts the most likely label) invariably expands the effective coverage of the expert pool. We further demonstrate the validity of this condition in broader stochastic settings in Appendix A. Serving as a sufficient condition for the consistency of PiCCE, this assumption leads to the following theorem:

**Theorem 5.2** (Consistency and Expert Accuracy Estimator)**.** *When Condition 1 holds and* (12) *and* (13) *is used as multiclass loss $\phi$ in* (9), *for any $\boldsymbol{x} \in \mathcal{X}$ and $\boldsymbol{\theta}^*$ minimizes* (10). *the prediction link $\varphi$ defined in* (5) *and $\boldsymbol{\theta}^*$ reproduces the Bayes optimal decision $f^*(\boldsymbol{x})$, i.e.: $\varphi(\boldsymbol{\theta}^*) = f^*(\boldsymbol{x})$, and $\mathrm{Argmax}_{j\in[J]}\theta^*_{j+K} = \mathrm{Argmax}_{j\in[J]}\mathrm{Acc}_j(\boldsymbol{x}) = \{j^*\}$. Furthermore:*

*(A). When* (12) *is used, $u^*_{j^*} = \mathrm{Acc}_{j^*}(\boldsymbol{x})\widetilde{V}(\boldsymbol{x})$.*

*(B). When* (13) *is used, $s(\theta^*_{j^*}) = \mathrm{Acc}_{j^*}(\boldsymbol{x})$.*

This theorem formally validates the consistency of the L2D system driven by PiCCE. Beyond system-level consistency, it facilitates the consistent estimation of the optimal expert's accuracy, which is a vital component for uncertainty quantification (Verma & Nalisnick, 2022). Notably, the OvA log loss formulation allows PiCCE to natively extract expert accuracy via $\mathrm{argmax}_{j\in[J]}s(\theta_{K+j})$ without extra post-processing. This aligns with the insights of (Verma & Nalisnick, 2022; Cao et al., 2023), which highlighted the efficacy of OvA-based methods in modeling expert performance. Finally, we remark that Condition 1 is merely a sufficient condition, not a necessary one. This suggests that the theoretical guarantees of PiCCE likely extend to scenarios exceeding these assumptions, pointing towards a promising direction for future exploration.

## 6  Experiments

In this section, we evaluate PiCCE on both synthetic and real-world expert settings to validate our theoretical findings and empirical effectiveness in multi-expert L2D. Additional experimental details and results are provided in Appendix D.

### 6.1  Experimental Setup

**Models and Datasets**  For synthetic experts, we conduct experiments on CIFAR-100 (Krizhevsky et al., 2009) and ImageNet (Deng et al., 2009). For ImageNet, we consider a 120-class dog subset and construct synthetic domain-

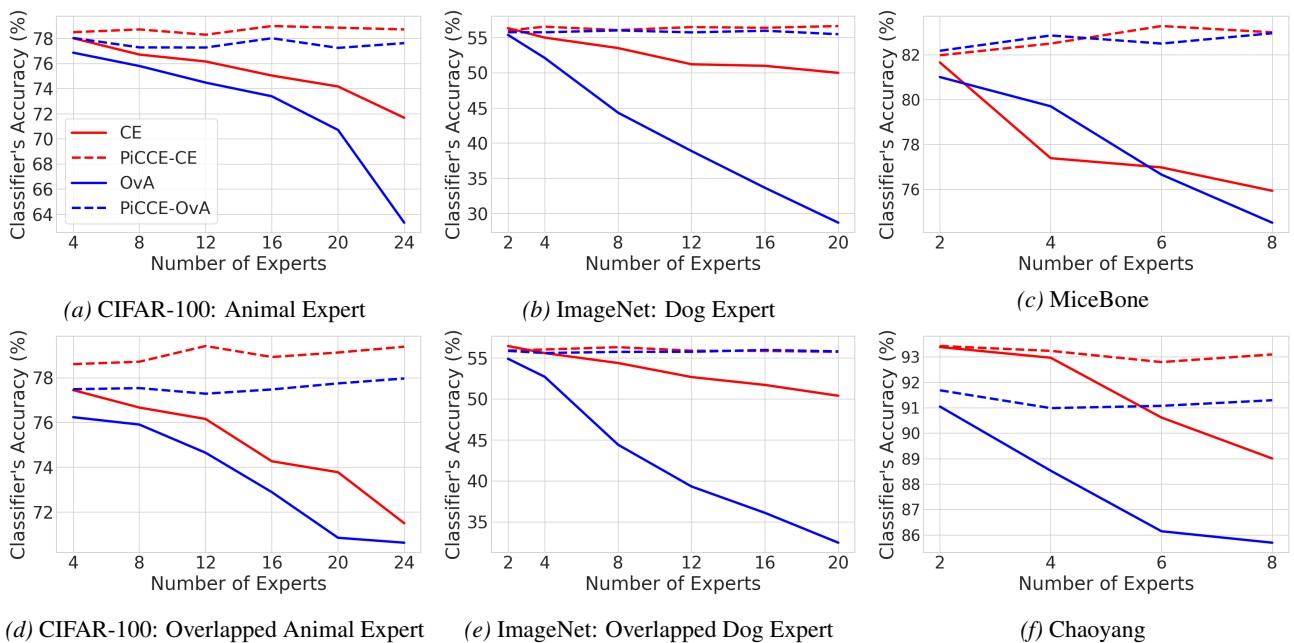

*Figure 2.* Classifier accuracy vs. number of experts on synthetic (CIFAR-100, ImageNet) and real-world expert datasets (MiceBone, Chaoyang). Solid lines denote methods derived from (4), while dashed lines correspond to those derived from (9). Across all datasets, existing methods exhibit a performance drop as the number of experts increases, whereas PiCCE remains stable.

specialized dog experts under several controlled settings, including disjoint domains, overlapped domains, and varying accuracies. For CIFAR-100, a similar expert construction is adopted for biological-domain experts over 50 biological classes. Details are provided in Appendix D. Following previous works (Verma & Nalisnick, 2022; Narasimhan et al., 2022), we use a 28-layer WideResNet (Zagoruyko & Komodakis, 2016) and MobileNet-v2 (Sandler et al., 2018) as base models for CIFAR-100 and for ImageNet, respectively. We further consider two datasets with real-world human experts, MiceBone (Schmarje et al., 2022) and Chaoyang (Zhu et al., 2022), to demonstrate the practical effectiveness of PiCCE. We use ResNet-18 as base model for both datasets, with additional details provided in Appendix D.

**Baselines** Our evaluation focuses on jointly training methods, as post-hoc approaches require additional retraining. We evaluate combinations of the existing formulation (4) with cross-entropy and one-vs-all base losses, which correspond to the multi-expert CE and OvA surrogates. Our PiCCE method (9) instantiates the same base losses for a direct comparison, denoted as PiCCE-CE and PiCCE-OvA, respectively.

### 6.2 Experimental Results

**System's Accuracy and Coverage** In Table 1, we report the system error w.r.t. the target system loss (1) and coverage, defined as the ratio of non-deferred samples, under two different expert patterns (disjoint/overlapped) using the

multi-expert CE and OvA surrogate losses. The best results are highlighted in boldface. Across all settings, PiCCE consistently outperforms the vanilla CE and OvA formulations, achieving lower system error while maintaining higher coverage. Moreover, this performance improvement becomes more pronounced as the number of experts increases, which aligns with our theoretical analysis showing that existing surrogate formulations (4) can suffer from more severe underfitting with a larger expert aggregation term. Overall, these results demonstrate that PiCCE yields more robust system-level performance in multi-expert L2D settings. Results for the varying-accuracy expert pattern, which exhibit consistent trends, are provided in Appendix D.

In Table 2, we further provide the target system loss and coverage results on datasets with real-world experts, i.e., MiceBone and Chaoyang. Consistent with the synthetic results, PiCCE achieves improved system error and higher coverage across different numbers of experts, indicating that our proposed approach remains effective in practical settings with real-world human experts.

**Classifier's Accuracy** From Figure 2, we observe that under both synthetic expert settings (CIFAR-100 and ImageNet), as well as real-world experts (MiceBone and Chaoyang), the classifier prediction accuracy of CE and OvA (solid lines) consistently degrades as the number of experts increases. This degradation becomes increasingly pronounced with more experts and is particularly severe for OvA-based formulations, whose classifier accuracy drops

sharply across all expert settings. In contrast, methods derived from our PiCCE formulation (dashed lines) remain insensitive to the size of expert-set in the L2D system, maintaining consistently higher classifier accuracy across both synthetic and real-world datasets. These consistent trends across diverse settings indicate that the underfitting observed in existing multi-expert surrogate losses (4) is driven by the expert aggregation term, and that PiCCE effectively mitigates this underfitting issue by leveraging the empirical information to avoid such aggregation.

## 7  Conclusion

In this work, we have provided both empirical evidence and theoretical insights into the inherent classifier underfitting challenges of multi-expert L2D. We show that while underfitting in single-expert L2D requires additional assumptions, it arises naturally and inherently in the multi-expert setting due to the expert aggregation term. This fundamental difference renders existing L2D remedies ineffective. We address this issue by proposing PiCCE, a surrogate-based formulation that bypasses expert aggregation by regulating selection through ground-truth information. Our theoretical analysis and extensive experiments across synthetic and real-world benchmarks show that PiCCE effectively resolves multi-expert underfitting and consistently outperforms state-of-the-art methods.

## Acknowledgements

The project is partially supported by National Research Foundation, Singapore, under the NRF RSS Scheme NRF-RSS2022-009 and the Singapore Global AI VP grant AIVP-2024-002. Yuzhou Cao is supported by Google PhD Fellowship program.

## Impact Statement

This paper presents work whose goal is to advance the field of Machine Learning. There are many potential societal consequences of our work, none which we feel must be specifically highlighted here.

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

## A    Condition 1 with Stochastic Experts

The stochastic expert setting is more complex compared with the deterministic expert cases and learning with rejection tasks (Chow, 1970; Bartlett & Wegkamp, 2008; Cortes et al., 2016; Charoenphakdee et al., 2021; Cao et al., 2022; 2023; Mao et al., 2023b; Narasimhan et al., 2024a;b; Cao et al., 2026), where the former assumes invariant experts with fixed decision rules and the latter assumes constant rejection costs. We show in this section that Condition 1 holds in broader stochastic settings, by giving the following examples:

**Example 1** (Dominant Expert). *Suppose $j^*$ is a dominant expert that its conditional accuracy equals or is higher than other experts on all classes, i.e., $\Pr(M_{j^*} = Y|Y = y, X = \boldsymbol{x}) \geq \Pr(M_j = Y|Y = y, X = \boldsymbol{x})$, and the inequality holds strictly on at least one class $y$. Furthermore, the expert predictions are conditional independent given label $Y$ (which is a commonly used condition in previous multi-expert L2D studies (Verma et al., 2023)). Then we can learn for any subset $\mathcal{M}$ and $j \notin \mathcal{M}$:*

$$\Pr(C_{\mathcal{M} \cup \{j\}}|X = \boldsymbol{x}) = \sum_{y=1}^{K} \eta_y(\boldsymbol{x})\Pr(C_{\mathcal{M} \cup \{j\}}||Y = y, X = \boldsymbol{x})$$

$$= \sum_{y=1}^{K} \eta_y(\boldsymbol{x}) \left(\Pr(C_{\mathcal{M}}|Y = y, X = \boldsymbol{x}) + \Pr(M_j = Y|Y = y, X = \boldsymbol{x}) - \Pr(C_{\mathcal{M}}|Y = y, X = \boldsymbol{x}) * \Pr(M_j = Y|Y = y, X = \boldsymbol{x})\right)$$

$$= \sum_{y=1}^{K} \eta_y(\boldsymbol{x}) \left(\Pr(C_{\mathcal{M}}|Y = y, X = \boldsymbol{x}) + \Pr(M_j = Y|Y = y, X = \boldsymbol{x}) \underbrace{(1 - \Pr(C_{\mathcal{M}}|Y = y, X = \boldsymbol{x}))}_{>0 \text{ if } \mathcal{M} \text{ does not include } j^*.}\right)$$

*Then for any $j' \neq j^*$:*

$$\Pr(C_{\mathcal{M} \cup \{j^*\}}|X = \boldsymbol{x}) - \Pr(C_{\mathcal{M} \cup \{j\}}|X = \boldsymbol{x})$$

$$= \sum_{y=1}^{K} \eta_y(\boldsymbol{x}) \left((\Pr(M_{j^*} = Y|Y = y, X = \boldsymbol{x}) - \Pr(M_j = Y|Y = y, X = \boldsymbol{x}))(1 - \Pr(C_{\mathcal{M}}|Y = y, X = \boldsymbol{x}))\right)$$

$$> 0,$$

*and thus Condition 1 holds.*

This example shows that Condition 1 can hold when expert predictions have randomness, where a dominant expert outperforms others across all classes. Furthermore, Condition 1 can be extended to milder scenarios. The following example demonstrates that the condition remains valid even when the expert is not globally superior, in which we focus on 3-class classification case for better presentation:

**Example 2** (Expert Only Dominant at the Major Class). *Suppose the class number is 3, and the the class conditional distribution and expert class-conditional accuracy are as below:*

| Class $y$ | $\eta_y(x)$ | $\Pr(M_1 = Y|Y = y, X = \boldsymbol{x})$ | $\Pr(M_2 = Y|Y = y, X = \boldsymbol{x})$ | $\Pr(M_3 = Y|Y = y, X = \boldsymbol{x})$ |
|---|---|---|---|---|
| $y$ =1 *(Major)* | **0.80** | **0.90** | 0.60 | 0.60 |
| $y$ =2 *(Minor)* | 0.10 | 0.40 | **0.90** | **0.90** |
| $y$ =3 *(Minor)* | 0.10 | 0.40 | **0.90** | **0.90** |

*Here we can see that $\mathrm{Acc}_1(\boldsymbol{x}) = 0.80 > \mathrm{Acc}_2(\boldsymbol{x}) = \mathrm{Acc}_3(\boldsymbol{x}) = 0.66$, and thus $j^* = 1$. However, we can learn the conditional accuracy of expert $M_1$ is lower than those of $M_2$ and $M_3$, thereby Example 1 cannot be applied here. However, we can infer Condition 1 still holds here. When $j = 2$ (the conclusion below holds symmetrically for $j = 3$ since $M_2$ and*

$M_3$ *has the same distribution)*:

$$\Pr(C_{\mathcal{M}\cup\{j^*\}}|X=\boldsymbol{x}) - \Pr(C_{\mathcal{M}\cup\{j\}}|X=\boldsymbol{x})$$

$$= \sum_{y=1}^{K} \eta_y(\boldsymbol{x}) \left( (\Pr(M_{j^*}=Y|Y=y,X=\boldsymbol{x}) - \Pr(M_j=Y|Y=y,X=\boldsymbol{x}))(1-\Pr(C_{\mathcal{M}}|Y=y,X=\boldsymbol{x})) \right)$$

$$= \sum_{y=1}^{K} \eta_y(\boldsymbol{x}) \left( (\Pr(M_1=Y|Y=y,X=\boldsymbol{x}) - \Pr(M_2=Y|Y=y,X=\boldsymbol{x}))(1-\Pr(M_3=Y|Y=y,X=\boldsymbol{x})) \right)$$

$$= 0.8*(0.9-0.6)(1-0.6) + 0.1*(0.4-0.9)(1-0.9) + 0.1*(0.4-0.9)(1-0.9)$$

$$= 0.8*0.3*0.4 - 0.1*0.5*0.1*2 = 0.086 > 0,$$

*and thus Condition 1 holds.*

This example reveals that Condition 1 covers a broad range of scenarios where expert stochasticity interacts with the data distribution. Intuitively, the optimal expert only needs to outperform others on the majority classes, while being allowed to be less accurate on rare classes.

These examples underscore the generality of Condition 1. A promising direction for future work is to further explore the necessary and sufficient conditions for Condition 1 to hold.

# B  Proofs of Conclusions in Section 4

## B.1  Proof of Theorem 4.2

*Proof.* For a given $\boldsymbol{\theta}_0 \in \mathbb{R}^{K+J}$, $\forall \epsilon > 0$, $\exists \sigma > 0$, such that $\forall \boldsymbol{\theta} \in B(\boldsymbol{\theta}_0, \sigma) = \{\boldsymbol{v} \in \mathbb{R}^{K+J} : \|\boldsymbol{\theta}_0 - \boldsymbol{v}\| < \sigma\}$:

$$|\ell_\phi(\boldsymbol{\theta}, y, \boldsymbol{m}) - \ell_\phi(\boldsymbol{\theta}_0, y, \boldsymbol{m})| = \left| \phi(\boldsymbol{\theta}, y) + \phi(\boldsymbol{\theta}, \operatorname*{argmax}_{j\in[\boldsymbol{m}]=y} \widetilde{\boldsymbol{\theta}}_j + K) - \left( \phi(\boldsymbol{\theta}_0, y) + \phi(\boldsymbol{\theta}_0, \operatorname*{argmax}_{j\in[\boldsymbol{m}=y]} \widetilde{\boldsymbol{\theta}}_{0j} + K) \right) \right|$$

$$= |\phi(\boldsymbol{\theta}, y) + \phi(\boldsymbol{\theta}, j^* + K) - (\phi(\boldsymbol{\theta}_0, y) + \phi(\boldsymbol{\theta}_0, j_0^* + K))|$$

$$= |(\phi(\boldsymbol{\theta}, y) - \phi(\boldsymbol{\theta}_0, y)) + (\phi(\boldsymbol{\theta}, j^* + K) - \phi(\boldsymbol{\theta}_0, j_0^* + K))|$$

where $j^*$ and $j_0^*$ denote the best expert under $\boldsymbol{\theta}$ and $\boldsymbol{\theta}_0$, respectively. We then complete the proof by considering the following two cases.

**Case 1:** $j_0^*$ is unique. Denote suboptimal solution by $\widetilde{J}_0^* := \operatorname*{argmax}_{j\in[\boldsymbol{m}=y]/\{j_0^*\}} \widetilde{\boldsymbol{\theta}}_{0j}$. For any $\widetilde{j}_0^* \in \widetilde{J}_0^*$, we assume that $\Delta := \|\boldsymbol{\theta}_{0j^*} - \boldsymbol{\theta}_{0\widetilde{j}^*}\|_2$, where $\|\cdot\|_2$ denotes the $L_2$ norm. Then by taking $\delta = \Delta/2$, we can obtain that $j_0^* = j^*$ and

$$|\ell_\phi(\boldsymbol{\theta}, y, \boldsymbol{m}) - \ell_\phi(\boldsymbol{\theta}_0, y, \boldsymbol{m})| = |(\phi(\boldsymbol{\theta}, y) - \phi(\boldsymbol{\theta}_0, y)) + (\phi(\boldsymbol{\theta}, j_0^* + K) - \phi(\boldsymbol{\theta}_0, j_0^* + K))|$$

$$\leq |\phi(\boldsymbol{\theta}, y) - \phi(\boldsymbol{\theta}_0, y)| + |\phi(\boldsymbol{\theta}, j_0^* + K) - \phi(\boldsymbol{\theta}_0, j_0^* + K)| \leq \epsilon$$

The last inequality holds since $\phi$ is continuous at $\boldsymbol{\theta}_0$.

**Case 2:** $j_0^*$ is not unique and belongs to $J_0^* := \operatorname*{argmax}_{j\in[\boldsymbol{m}=y]} \widetilde{\boldsymbol{\theta}}_{0j}$. Denote suboptimal solution by $\widetilde{J}_0^* := \operatorname*{argmax}_{j\in[\boldsymbol{m}=y]/J_0^*} \widetilde{\boldsymbol{\theta}}_{0j}$. For any $\widetilde{j}_0^* \in \widetilde{J}_0^*$, we assume that $\Delta := \|\boldsymbol{\theta}_{0j^*} - \boldsymbol{\theta}_{0\widetilde{j}^*}\|$, then for $\delta = \Delta/2$, we can obtain $j^* \in J_0^*$ and

$$|\ell_\phi(\boldsymbol{\theta}, y, \boldsymbol{m}) - \ell_\phi(\boldsymbol{\theta}_0, y, \boldsymbol{m})| = |(\phi(\boldsymbol{\theta}, y) - \phi(\boldsymbol{\theta}_0, y)) + (\phi(\boldsymbol{\theta}, j^* + K) - \phi(\boldsymbol{\theta}_0, j_0^* + K))|$$

$$\leq |\phi(\boldsymbol{\theta}, y) - \phi(\boldsymbol{\theta}_0, y)| + |\phi(\boldsymbol{\theta}, j^* + K) - \phi(\boldsymbol{\theta}_0, j_0^* + K)|$$

$$= |\phi(\boldsymbol{\theta}, y) - \phi(\boldsymbol{\theta}_0, y)| + |\phi(\boldsymbol{\theta}, j^* + K) - \phi(\boldsymbol{\theta}_0, j^* + K) + \phi(\boldsymbol{\theta}_0, j^* + K) - \phi(\boldsymbol{\theta}_0, j_0^* + K)|$$

$$\leq |\phi(\boldsymbol{\theta}, y) - \phi(\boldsymbol{\theta}_0, y)| + |\phi(\boldsymbol{\theta}, j^* + K) - \phi(\boldsymbol{\theta}_0, j^* + K)| + |\phi(\boldsymbol{\theta}_0, j^* + K) - \phi(\boldsymbol{\theta}_0, j_0^* + K)|$$

Because $\phi$ is symmetric w.r.t. its last J inputs and $\widetilde{\boldsymbol{\theta}}_{0j^*} = \widetilde{\boldsymbol{\theta}}_{0j_0^*}$, swapping these two coordinates does not change the input to $\phi$. Therefore, $|\phi(\boldsymbol{\theta}_0, j^* + K) - \phi(\boldsymbol{\theta}_0, j_0^* + K)| = 0$. Thus we have

$$|\ell_\phi(\boldsymbol{\theta}, y, \boldsymbol{m}) - \ell_\phi(\boldsymbol{\theta}_0, y, \boldsymbol{m})| \leq |\phi(\boldsymbol{\theta}, y) - \phi(\boldsymbol{\theta}_0, y)| + |\phi(\boldsymbol{\theta}, j^* + K) - \phi(\boldsymbol{\theta}_0, j^* + K)| \leq \epsilon$$

The last inequality holds since $\phi$ is continuous at $\boldsymbol{\theta}_0$.

Combing the conclusions above, we conclude the proof. $\qquad\square$

## B.2 Proof of Theorem 4.3

*Proof.* Denote by $\Sigma(\boldsymbol{\theta}) = \{\boldsymbol{\sigma} : \widetilde{\theta}_{\sigma_1} \geq \cdots \geq \widetilde{\theta}_{\sigma_J}\}$ the set of all descending permutations of $\boldsymbol{\theta}$, we then obtain that:

$$R_{\ell_\phi|X=\boldsymbol{x}}(\boldsymbol{\theta}) = \mathbb{E}_{Y,\boldsymbol{M}|X=x}[\phi(\boldsymbol{\theta}, Y) + \phi(\boldsymbol{\theta}, \operatorname*{argmax}_{j\in[\boldsymbol{M}=Y]} \widetilde{\theta}_j + K)]$$

$$= \sum_{y=1}^{K} \eta_y \phi(\boldsymbol{\theta}, y) + \sum_{\boldsymbol{\sigma}\in\Sigma(\boldsymbol{\theta})} \Pr(\boldsymbol{\sigma})\Big(\Pr(M_{\sigma_1} = Y \mid X = \boldsymbol{x})\phi(\boldsymbol{\theta}, \sigma_1 + K) + \Pr(M_{\sigma_1} \neq Y, M_{\sigma_2} = Y \mid X = \boldsymbol{x})\phi(\boldsymbol{\theta}, \sigma_2 + K)$$

$$+ \cdots + \Pr(M_{\sigma_{1:J-1}} \neq Y, M_{\sigma_J} = Y \mid X = \boldsymbol{x})\phi(\boldsymbol{\theta}, \sigma_J + K)\Big)$$

$$= \sum_{y=1}^{K} \eta_y \phi(\boldsymbol{\theta}, y) + \sum_{\boldsymbol{\sigma}\in\Sigma(\boldsymbol{\theta})} \Pr(\boldsymbol{\sigma}) \left(\sum_{j=1}^{J} \Pr(M_{\sigma_{1:j-1}} \neq Y, M_{\sigma_j} = Y|X = x)\phi(\boldsymbol{\theta}, \sigma_j + K)\right)$$

Since $\phi$ is symmetric to the last $J$ inputs, for any $\boldsymbol{\sigma} \in \Sigma(\boldsymbol{\theta})$, there exists a permutation matrix $\widetilde{P} \in \mathbb{R}^{J\times J}$ such that

$$\sum_{j=1}^{J} \Pr(M_{\sigma_{1:j-1}} \neq Y, M_{\sigma_j} = Y|X = x)\phi(\boldsymbol{\theta}, \sigma_j + K)$$

$$= \Big[\Pr(M_{\sigma_1} = Y)|X = \boldsymbol{x}), \cdots, \Pr(M_{\sigma_{1:J-1}} \neq Y, M_{\sigma_J} = Y|X = x)\Big]\Big[\phi(\boldsymbol{\theta}, \sigma_1 + K), \cdots, \phi(\boldsymbol{\theta}, \sigma_J + K)\Big]^{\top}$$

$$= \Big[\Pr(M_{\sigma_1} = Y)|X = \boldsymbol{x}), \cdots, \Pr(M_{\sigma_{1:J-1}} \neq Y, M_{\sigma_J} = Y|X = x)\Big]\widetilde{P}\boldsymbol{\phi}(\widetilde{\boldsymbol{\theta}})$$

$$= \Big[\Pr(M_1 = Y, \bigcap_{j\in\{i\in[J]:\widetilde{\theta}_i\geq\widetilde{\theta}_1\}} M_j \neq Y|X = \boldsymbol{x}), \cdots, \Pr(M_J = Y, \bigcap_{j\in\{i\in[J]:\widetilde{\theta}_i\geq\widetilde{\theta}_J\}} M_j \neq Y|X = \boldsymbol{x})\Big]\boldsymbol{\phi}(\widetilde{\boldsymbol{\theta}})$$

where $\boldsymbol{\phi}(\widetilde{\boldsymbol{\theta}}) = [\phi(\boldsymbol{\theta}, 1 + K), \cdots, \phi(\boldsymbol{\theta}, J + K)]^{\top}$. Therefore,

$$\sum_{\boldsymbol{\sigma}\in\Sigma(\boldsymbol{\theta})} \Pr(\boldsymbol{\sigma}) \left(\sum_{j=1}^{J} \Pr(M_{\sigma_{1:j-1}} \neq Y, M_{\sigma_j} = Y \mid X = x)\phi(\boldsymbol{\theta}, \sigma_j + K)\right)$$

$$= \left(\sum_{\boldsymbol{\sigma}\in\Sigma(\boldsymbol{\theta})} \Pr(\boldsymbol{\sigma})\right)\Big[\Pr(M_1 = Y, \bigcap_{j\in\{i\in[J]:\widetilde{\theta}_i\geq\widetilde{\theta}_1\}} M_j \neq Y \mid X = \boldsymbol{x}), \cdots, \Pr(M_J = Y, \bigcap_{j\in\{i\in[J]:\widetilde{\theta}_i\geq\widetilde{\theta}_J\}} M_j \neq Y \mid X = \boldsymbol{x})\Big]\boldsymbol{\phi}(\widetilde{\boldsymbol{\theta}})$$

$$= \sum_{j=1}^{J} \Pr(M_{\sigma_{1:j-1}} \neq Y, M_{\sigma_j} = Y \mid X = x)\phi(\boldsymbol{\theta}, \sigma_j + K), \text{ for any } \boldsymbol{\sigma} \in \Sigma(\boldsymbol{\theta}),$$

which completes the proof. $\qquad\square$

## B.3 Proof of Theorem 4.4

*Proof.*

$$\sum_{y=1}^{K} \eta_y + \sum_{j=1}^{J} \Pr(M_{\sigma_{1:j-1}} \neq Y, M_{\sigma_j} = Y \mid X = \boldsymbol{x}) = 1 + \sum_{j=1}^{J}\Pr(\bigcap_{i<j}(M_{\sigma_i} = Y)^c \cap (M_{\sigma_j} = Y) \mid X = \boldsymbol{x})$$

$$= 1 + \Pr(\bigcup_{j\in[J]} M_{\sigma_j} = Y \mid X = \boldsymbol{x})$$

$$= 1 + \Pr(\bigcup_{j\in[J]} M_j = Y \mid X = \boldsymbol{x})$$

where $(M_{\sigma_i} = Y)^c$ denotes the complement event of $M_{\sigma_i} = Y$. $\square$

## C Proof of Conclusions in Section 5

In this proof, we focus on a fixed but arbitrary $x$, and the derivation can be extended to any $x \in \mathcal{X}$, and thus we omit the dependence of $\eta$ on $x$.

### C.1 Proof of Lemma 5.1

C.1.1 PROOF OF LEMMA 5.1, (A).

*Proof.* Denote by $\widehat{\eta}_y = \text{softmax}(\boldsymbol{\theta})_y$ and $u_j = \text{softmax}(\boldsymbol{\theta})_{K+j}$. According to the risk formulation in Theorem 4.3, the optimization problem can be formulated as below:

$$\min_{\widehat{\boldsymbol{\eta}}, \boldsymbol{u}} \ -\sum_{y=1}^{K} \eta_y \log \widehat{\eta}_y - \sum_{j=1}^{J} \Pr(M_{\sigma_{1:j-1}} \neq Y, M_{\sigma_j} = Y \mid X = \boldsymbol{x}) \log u_{\sigma_j}.$$
$$\text{s.t.} \ \ \widehat{\eta}_y, u_j \in [0,1], \ \forall y \in [K], j \in [J]$$
$$\sum_{i=1}^{K} \widehat{\eta}_i + \sum_{j=1}^{J} u_j = 1.$$

**Regularity of this problem:** First of all, we can notice that **the minimizer is attainable** according to the continuity of the objective (Theorem 4.2) and that the feasible region is closed.

Denote by the optimal classifier $\widehat{\boldsymbol{\eta}}^*$ and optimal expert score $\boldsymbol{u}^*$. we can first notice that $\sum_{j=1}^{J} u_j^* < 1$, otherwise the value of the objective will be infinitely large since $\widehat{\boldsymbol{\eta}} = 0$. Then we continue the proof:

- **Step 1:** For any fixed $\boldsymbol{u}$, we can learn that $\widehat{\eta}_y = \eta_y(1 - \sum_{j=1}^{J} u_j)$.

  In this case, we write $\widehat{\boldsymbol{\eta}}$ as a function of $\boldsymbol{u}$. Then we can omit the constant $\boldsymbol{\eta}$ and the problem can be formulated as:

$$\min_{\boldsymbol{u}} \ F(\boldsymbol{u}) = a - \log(1 - \sum_{j=1}^{J} u_j) - \sum_{j=1}^{J} \Pr(M_{\sigma_{1:j-1}} \neq Y, M_{\sigma_j} = Y \mid X = \boldsymbol{x}) \log u_{\sigma_j}.$$
$$\text{s.t.} \ \ u_j \in [0,1], \ \forall y \in [K], j \in [J].$$
$$\sum_{j=1}^{J} u_j < 1.$$

- **Step 2:** According to the regularity, suppose the minimizer of this problem $\boldsymbol{u}^*$. Then we prove that $\widetilde{V}(\boldsymbol{x}) = \frac{V(\boldsymbol{x})}{1+V(\boldsymbol{x})}$ by contradiction. Suppose $\sum_{j=1}^{J} u_j = \frac{V(\boldsymbol{x})}{1+V(\boldsymbol{x})} + \delta$, where $\delta \neq 0$ and $\frac{V(\boldsymbol{x})}{1+V(\boldsymbol{x})} + \delta \in [0,1)$. Then denote by

$\widetilde{u} = \left( \frac{\frac{V(x)}{1+V(x)}}{\frac{V(x)}{1+V(x)} + \delta} \right) u$, and we can then learn:

$$
\begin{aligned}
F(u) - F(\widetilde{u}) = & -\log\left(1 - \frac{V(x)}{1+V(x)} - \delta\right) - \sum_{j=1}^{J} \Pr(M_{\sigma_{1:j-1}} \neq Y, M_{\sigma_j} = Y \mid X = x) \log u_{\sigma_j} \\
& + \log\left(1 - \frac{V(x)}{1+V(x)}\right) + \sum_{j=1}^{J} \Pr(M_{\sigma_{1:j-1}} \neq Y, M_{\sigma_j} = Y \mid X = x) \log u_{\sigma_j} \\
& + \underbrace{\sum_{j=1}^{J} \Pr(M_{\sigma_{1:j-1}} \neq Y, M_{\sigma_j} = Y \mid X = x)}_{= V(x) \text{ according to Lemma 4.4}} \left(\log \frac{V(x)}{1+V(x)} - \log\left(\frac{V(x)}{1+V(x)} + \delta\right)\right) \\
= & \left(-\log\left(1 - \frac{V(x)}{1+V(x)} - \delta\right) - V(x)\log\left(\frac{V(x)}{1+V(x)} + \delta\right)\right) \\
& - \left(-\log\left(1 - \frac{V(x)}{1+V(x)}\right) - V(x)\log\left(\frac{V(x)}{1+V(x)}\right)\right) \\
> & \ 0
\end{aligned}
$$

The last inequality is obtained since log loss is a proper loss (Gneiting & Raftery, 2007; Reid & Williamson, 2010; Williamson et al., 2016; Bao, 2023; Bao & Takatsu, 2025), and thus $F(u) > F(\widetilde{u})$, which indicates that $\widetilde{V}(x) = \frac{V(x)}{1+V(x)}$.

Combining Step 1 and Step 2 and we can conclude the proof. □

### C.1.2 PROOF OF LEMMA 5.1, (B).

*Proof.* For any $\theta \in \mathbb{R}^{K+j}$, we abuse the notation $s_i := s(\theta_i)$, which omits the dependence on $\theta$. Furthermore, we denote by $u_j := s_{j+K}$. According to the risk formulation in Theorem 4.3, the risk minimization problem can be formulated as:

$$
\begin{aligned}
\min_{s,u} \ & \underbrace{-\sum_{y=1}^{K} \eta_y \log s_y - \sum_{y=1}^{K} (1-\eta_y) \log(1-s_y)}_{F(s_{1:K})} - \sum_{j=1}^{J} \Pr(M_{\sigma_{1:j-1}} \neq Y, M_{\sigma_j} = Y \mid X = x) \log u_{\sigma_j} \\
& - \sum_{j=1}^{J} (1 - \Pr(M_{\sigma_{1:j-1}} \neq Y, M_{\sigma_j} = Y \mid X = x)) \log(1 - u_{\sigma_j}) \\
s.t. \ & s_i, u_j \in [0,1], \ \forall i \in [K], j \in [J].
\end{aligned}
$$

First of all, the minimizer is attainable since the feasible region is closed and the target is continuous. Furthermore, the optimization problem is separable since $s_{1:K}$ and $s_{K+1:K+J}$ are independent in both the target and feasible region. Notice that $F(s_{1:K})$ is minimized at $s_y = \eta_y$ since $s_y \in [0,1]$, which concludes the proof. □

### C.2 Proof of Theorem 5.2

### C.2.1 PROOF OF THEOREM 5.2, (A).

*Proof.* Denote by $\widehat{\eta}_y = \text{softmax}(\theta)_y$, $u_j = \text{softmax}(\theta)_{K+j}$, $\mathcal{S}$ the set that $u_j \in [0,1]$, and $\sum_{j \in [J]} u_j = \widetilde{V}(x)$.

- **Step 1: First we prove that** $j^* \in \text{Argmax}_{j \in [J]} u_j^*$. We prove it by contradiction. Assuming $j^* \notin \text{Argmax}_{j \in [J]} u_j^*$, then we can find a permutation $\sigma$ that $u_{\sigma_1}^* \geq \cdots \geq u_{\sigma_J}^*$ and $u_{\sigma_1}^* > u_{j^*}^*$. According to Lemma 5.1, the risk can be written as:

$$-\sum_{y=1}^{K}\eta_y\log\eta_y(1-\widetilde{V}(\boldsymbol{x}))-\underbrace{\sum_{j=1}^{J}\Pr(M_{\sigma_{1:j-1}}\neq Y,M_{\sigma_j}=Y|X=\boldsymbol{x})\log u^*_{\sigma_j}}_{P(\boldsymbol{u}^*)},$$

and we focus on $P(\boldsymbol{u}^*)$ since the first part is unchanged for any $\boldsymbol{u}\in\mathcal{S}$. Denote by $\boldsymbol{u}'$ that $u'_{\sigma_1}=u^*_{j^*}$, $u'_{j^*}=u^*_{\sigma_1}$, and $u'_j=u^*_j$ for $j\notin\{\sigma_1,j^*\}$. Denote by $j'$ the element that $\sigma_{j'}=j^*$. Then $u'_{\sigma'_1}\geq\cdots\geq u'_{\sigma'_J}$ for $\boldsymbol{\sigma}'$ that $\sigma'_1=j^*$, $\sigma'_{j'}=\sigma_1$, and $\sigma'_j=\sigma_j$ for $j\notin\{1,j'\}$.

Thus $P(\boldsymbol{u}')$ can be written as:

$$P(\boldsymbol{u}')=-\sum_{j=1}^{J}\Pr(M_{\sigma'_{1:j-1}}\neq Y,M_{\sigma'_j}=Y|X=\boldsymbol{x})\log u'_{\sigma'_j}$$

$$=-\sum_{j=1}^{J}\Pr(M_{\sigma'_{1:j-1}}\neq Y,M_{\sigma'_j}=Y|X=\boldsymbol{x})\log u^*_{\sigma_j}$$

Denote by $T(\boldsymbol{p})=-\sum_{j=1}^{J}p_j\log u^*_{\sigma_j}$. Also denote by $\boldsymbol{p}^*=[\Pr(M_{\sigma_{1:j-1}}\neq Y,M_{\sigma_j}=Y|X=\boldsymbol{x})]_{j=1}^{J}$ and $\boldsymbol{p}'=[\Pr(M_{\sigma'_{1:j-1}}\neq Y,M_{\sigma'_j}=Y|X=\boldsymbol{x})]_{j=1}^{J}$, we can learn $T(\boldsymbol{p}^*)=P(\boldsymbol{u}^*)$ and $T(\boldsymbol{p}')=P(\boldsymbol{u}')$.

Denote by $S^*_j=\sum_{j'=1}^{j}p^*_{j'}=\Pr(C_{\sigma_{1:j}}|X=\boldsymbol{x})$ and $S'_j=\sum_{j'=1}^{j}p'_{j'}=\Pr(C_{\sigma'_{1:j}}|X=\boldsymbol{x})$. Also since $\boldsymbol{u}^*$ is not a constant vector, we can find $\tilde{j}$ that $u^*_{\sigma_{\tilde{j}}}>u^*_{\sigma_{\tilde{j}+1}}$. Then we can further decompose $P(\boldsymbol{u}^*)-P(\boldsymbol{u}')$ into:

$$P(\boldsymbol{u}^*)-P(\boldsymbol{u}')=T(\boldsymbol{p}^*)-T(\boldsymbol{p}')$$

$$=-\sum_{j=1}^{J}(p^*_j-p'_j)\log u^*_{\sigma_j}$$

$$=\sum_{j=1}^{J-1}(S^*_j-S'_j)(\log u^*_{\sigma_{j+1}}-\log u^*_{\sigma_j})\quad\text{(Summation by parts)}$$

$$=\sum_{j=1}^{J-1}\underbrace{(\Pr(C_{\sigma_{1:j}}|X=\boldsymbol{x})-\Pr(C_{\sigma'_{1:j}}|X=\boldsymbol{x}))}_{<0\text{ according to Condition 1}}\underbrace{(\log u^*_{\sigma_{j+1}}-\log u^*_{\sigma_j})}_{\leq 0\text{ since }u^*_{\sigma_j}\text{ is in descending order}}$$

$$\geq\underbrace{(\Pr(C_{\sigma_{1:\tilde{j}}}|X=\boldsymbol{x})-\Pr(C_{\sigma'_{1:\tilde{j}}}|X=\boldsymbol{x}))}_{<0\text{ according to Condition 1}}\underbrace{(\log u^*_{\sigma_{\tilde{j}+1}}-\log u^*_{\sigma_{\tilde{j}}})}_{<0}$$

$$>0.$$

Then we can learn $P(\boldsymbol{u}^*)>P(\boldsymbol{u}')$, which contradicts that $\boldsymbol{u}^*$ is the minimizer.

- **Step 2: Second we prove that** $\text{Argmax}_{j\in[J]}u^*_j=\{j^*\}$. Suppose $|\text{Argmax}_{j\in[J]}u^*_j|=J'>1$, Then there exists a permutation $\boldsymbol{\sigma}$ that $\sigma_1=j^*$, $u^*_{\sigma_1},\cdots,u^*_{\sigma_{J'}}=u^*_{j^*}$, and $u^*_{\sigma_{J'+1}}\geq\cdots u^*_{\sigma_J}$. Then $P(\boldsymbol{u}^*)$ can be written as:

$$P(\boldsymbol{u}^*)=-\sum_{j=1}^{J}\Pr(M_{\sigma_{1:j-1}}\neq Y,M_{\sigma_j}=Y|X=\boldsymbol{x})\log u^*_{\sigma_j}$$

$$=-\sum_{j=1}^{J'}\Pr(M_{\sigma_{1:j-1}}\neq Y,M_{\sigma_j}=Y|X=\boldsymbol{x})\log u^*_{j^*}$$

$$-\sum_{j=J'+1}^{J}\Pr(M_{\sigma_{1:j-1}}\neq Y,M_{\sigma_j}=Y|X=\boldsymbol{x})\log u^*_{\sigma_j}$$

Denote by $\delta > 0$ that $\delta < \min\left\{ \frac{u^*_{j^*} - u^*_{\sigma_{J'+1}}}{u^*_{j^*}}, \frac{1 - u^*}{u^*} \right\} = b$. Denote by $\boldsymbol{u}'$ that $u'_{\sigma_1} = (1 + \delta)u^*_{j^*}$, $u'_{\sigma_{J'}} = (1 - \delta)u^*_{j^*}$, and other elements equals those of $\boldsymbol{u}^*$. In this case, we can still get $\boldsymbol{u}'_{\sigma_1} \geq \cdots \geq \boldsymbol{u}'_{\sigma_1}$. Then $P(\boldsymbol{u}')$ can be written as:

$$P(\boldsymbol{u}') = -\sum_{j=1}^{J} \Pr(M_{\sigma_{1:j-1}} \neq Y, M_{\sigma_j} = Y | X = \boldsymbol{x}) \log u'_{\sigma_j}$$

$$= -\sum_{j=J'+1}^{J} \Pr(M_{\sigma_{1:j-1}} \neq Y, M_{\sigma_j} = Y | X = \boldsymbol{x}) \log u^*_{\sigma_j}$$

$$- \Pr(M_{\sigma_{1:J'-1}} \neq Y, M_{\sigma_{J'}} = Y | X = \boldsymbol{x}) \log(1 - \delta) u^*_{j^*}$$

$$- \Pr(M_{j^*} = Y | X = \boldsymbol{x}) \log(1 + \delta) u^*_{j^*}$$

Then:

$$P(\boldsymbol{u}^*) - P(\boldsymbol{u}') = \Pr(M_{\sigma_{1:J'-1}} \neq Y, M_{\sigma_{J'}} = Y | X = \boldsymbol{x}) \log(1 - \delta)$$
$$+ \Pr(M_{j^*} = Y | X = \boldsymbol{x}) \log(1 + \delta)$$

Notice $\Pr(M_{\sigma_{1:J'-1}} \neq Y, M_{\sigma_{J'}} = Y | X = \boldsymbol{x}) \leq \Pr(M_{\sigma_{J'}} = Y | X = \boldsymbol{x}) < \Pr(M_{j^*} = Y | X = \boldsymbol{x})$. Denote by $A := \Pr(M_{j^*} = Y | X = \boldsymbol{x})$, $B := \Pr(M_{\sigma_{1:J'-1}} \neq Y, M_{\sigma_{J'}} = Y | X = \boldsymbol{x})$. We can learn that for any $\delta \in \left(0, \min\left\{b, \frac{A-B}{A+B}\right\}\right)$:

$$P(\boldsymbol{u}^*) - P(\boldsymbol{u}') = \log\left((1 - \delta)^B (1 + \delta)^A\right) = F(\delta).$$

Notice that $F'(\delta) = \frac{A}{1+\delta} - \frac{B}{1-\delta} > 0$ on $\left(0, \frac{A-B}{A+B}\right)$, and $F(0) = 0$, then we can learn $F(\delta) > 0$ on $\left(0, \min\left\{b, \frac{A-B}{A+B}\right\}\right)$, and then $P(\boldsymbol{u}^*) > P(\boldsymbol{u}')$ for such $\boldsymbol{u}'$, which has unique argmax element $j^*$.

In conclusion, for any $\boldsymbol{u}^*$ with multiple argmax elements, we can always find on $\boldsymbol{u}'$ with unique argmax element $j^*$ that has a lower risk, which concludes the proof.

**Step 1 and Step 2 concludes that** $\text{Argmax}_{j \in [J]} u^*_j$ **is uniquely obtained at** $j^*$**.**

- **Step 3: Finally we prove** $u^*_{j^*} = \text{Acc}_{j^*}(\boldsymbol{x})(1 - \widetilde{V}(\boldsymbol{x}))$**.** According to Step 1, we have that $j^* \in \text{Argmax}_{j \in [J]} u^*_j$. Denote by $\mathcal{S}_\sigma := \{\boldsymbol{u} \in \mathcal{S} : u_{\sigma_1} \geq \cdots u_{\sigma_J}\}$, we can learn that $\boldsymbol{u}^* \in \mathcal{S}_\sigma$ for some $\sigma$ that $\sigma_1 = j^*$. Then we turn to prove that for any $\sigma$ that $\sigma_1 = j^*$, $u'_{j^*} = \text{Acc}_{j^*}(\boldsymbol{x})(1 - \widetilde{V}(\boldsymbol{x}))$ for any $\boldsymbol{u}' \in \text{Argmin}_{\boldsymbol{u} \in \mathcal{S}_\sigma} P(\boldsymbol{u})$.

Notice that $\sigma_1 = j^*$, and thus $\Pr(M_{\sigma_1} = Y | X = \boldsymbol{x}) > \Pr(M_{\sigma_{1:j-1}} \neq Y, M_{\sigma_j} = Y | X = \boldsymbol{x})$ for any $j > 1$. Then we formulate the optimization problem as below:

$$\min_{\boldsymbol{u}} -\sum_{j=1}^{J} \Pr(M_{\sigma_{1:j-1}} \neq Y, M_{\sigma_j} = Y | X = \boldsymbol{x}) \log u_{\sigma_j}$$

$$s.t. \quad \sum_{j=1}^{J} u_j = \widetilde{V}(\boldsymbol{x})$$

$$u_{\sigma_{j+1}} - u_{\sigma_j} \leq 0$$

Notice that the objective is convex and constraints are affine, and thus KKT conditions are sufficient and necessary. Then we conduct KKT analysis. For clear representation, let $\mu_{\sigma_0} = \mu_{\sigma_J} = 0$. The KKT conditions can be written as:

$$\begin{cases} \Pr(M_{\sigma_{1:j-1}} \neq Y, M_{\sigma_j} = Y | X = \boldsymbol{x}) = u_{\sigma_j}(\lambda + \mu_{\sigma_{j-1}} - \mu_{\sigma_j}), & \text{(Stationary condition)} \\ \mu_{\sigma_j} \geq 0, & \text{(Dual feasibility)} \\ \mu_{\sigma_j}(u_{\sigma_{j+1}} - u_{\sigma_j}) = 0. & \text{(Complementary slackness)} \end{cases}$$

Summing up the stationary condition for $j = 1, \cdots, J$ and we can learn:

$$V(\boldsymbol{x}) = \lambda\widetilde{V}(\boldsymbol{x}) + \sum_{j=1}^{J} u_{\sigma_j}(\mu_{\sigma_{j-1}} - \mu_{\sigma_j}) = \lambda\widetilde{V}(\boldsymbol{x}) + \underbrace{\sum_{j=1}^{J} \mu_{\sigma_j}(u_{\sigma_{j-1}} - u_{\sigma_j})}_{=0 \text{ according to complementary slackness}}$$

Then we can learn $\lambda = 1 + V(\boldsymbol{x})$. Again according to stationary condition we can learn:

$$\text{Acc}_{j^*}(\boldsymbol{x}) = u_{j^*}(1 + V(\boldsymbol{x}) - \mu_{j^*}).$$

Then we can learn $\mu_{j^*} = 1 + V(\boldsymbol{x}) - \frac{\text{Acc}_{j^*}(\boldsymbol{x})}{u_{j^*}}$. then we can learn $\mu_{j^*} = 0$ since $u_{\sigma_2} - u_{j^*} < 0$. and thus $u_{j^*} = \frac{\text{Acc}_{j^*}(\boldsymbol{x})}{1+V(\boldsymbol{x})} = \text{Acc}_{j^*}(\boldsymbol{x})(1 - \widetilde{V}(\boldsymbol{x}))$.

Then we can conclude the proof since softmax operator is order-preserving. □

### C.2.2 PROOF OF THEOREM 5.2, (B).

*Proof.* Since $\theta_j^*$ has been solved in Lemma 5.1, we focus on the following problem:

$$\min_{\boldsymbol{u} \in [0,1]^J} P(\boldsymbol{u}) = -\sum_{j=1}^{J} \Pr(M_{\sigma_{1:j-1}} \neq Y, M_{\sigma_j} = Y \mid X = \boldsymbol{x}) \log u_{\sigma_j}$$

$$-\sum_{j=1}^{J}(1 - \Pr(M_{\sigma_{1:j-1}} \neq Y, M_{\sigma_j} = Y \mid X = \boldsymbol{x})) \log(1 - u_{\sigma_j})$$

Denote by $\boldsymbol{u}^*$ any minimizer of $P(\boldsymbol{u})$, which exists according to Lemma 5.1:

- **Step 1:** We first prove that $j^* \in \text{Argmax}_{j \in [J]} u_j^*$.

  Denote by $\boldsymbol{\sigma}$ a descending permutation of $\boldsymbol{u}^*$. Suppose $\sigma_1, \cdots, \sigma_i \in \text{Argmax}_{j \in [J]} u_j^*$ and $\sigma_{i^*} = j^* \notin \text{Argmax}_{j \in [J]} u_j^*$. Then swapping $u_{\sigma_1}^*$ and $u_{j^*}^*$ and we construct $\boldsymbol{u}'$, which has a descending permutation $\boldsymbol{\sigma}'$ that $\sigma_1' = j^*$. Denote by $T(\boldsymbol{p}) = -\sum_{j=1}^{J} p_j \log u_{\sigma_j}^* - \sum_{j=1}^{J}(1 - p_j)\log(1 - u_{\sigma_j}^*)$. Also denote by $\boldsymbol{p}^* = [\Pr(M_{\sigma_{1:j-1}} \neq Y, M_{\sigma_j} = Y|X = \boldsymbol{x})]_{j=1}^{J}$, $\boldsymbol{p}' = [\Pr(M_{\sigma'_{1:j-1}} \neq Y, M_{\sigma'_j} = Y|X = \boldsymbol{x})]_{j=1}^{J}$, $S_j^* = \sum_{j'=1}^{j} p_{j'}^* = \Pr(C_{\sigma_{1:j}}|X = \boldsymbol{x})$, and $S_j' = \sum_{j'=1}^{j} p_{j'}' = \Pr(C_{\sigma'_{1:j}}|X = \boldsymbol{x})$, we can learn $P(\boldsymbol{u}^*) = T(\boldsymbol{p}^*)$ and $P(\boldsymbol{u}') = T(\boldsymbol{p}')$. We have the following condition:

$$P(\boldsymbol{u}^*) - P(\boldsymbol{u}') = T(\boldsymbol{p}^*) - T(\boldsymbol{p}') = -\sum_{j=1}^{J}(p_j^* - p_j')\log u_{\sigma_j}^* - \sum_{j=1}^{J}(p_j' - p_j^*)\log(1 - u_{\sigma_j}^*)$$

$$= \sum_{j=1}^{J-1}(S_j^* - S_j')(\log u_{\sigma_{j+1}}^* - \log u_{\sigma_j}^*) + \sum_{j=1}^{J-1}(S_j' - S_j^*)(\log(1 - u_{\sigma_{j+1}}^*) - \log(1 - u_{\sigma_j}^*)) \quad \text{(Summation by parts)}$$

$$= \sum_{j=1}^{J-1} \underbrace{(\Pr(C_{\sigma_{1:j}}|X = \boldsymbol{x}) - \Pr(C_{\sigma'_{1:j}}|X = \boldsymbol{x}))}_{<0 \text{ according to Condition } 1} \underbrace{(\log u_{\sigma_{j+1}}^* - \log u_{\sigma_j}^*)}_{\leq 0 \text{ since } u_{\sigma_j}^* \text{ is in descending order}}$$

$$+ \sum_{j=1}^{J-1} \underbrace{(\Pr(C_{\sigma'_{1:j}}|X = \boldsymbol{x}) - \Pr(C_{\sigma_{1:j}}|X = \boldsymbol{x}))}_{>0 \text{ according to Condition } 1} \underbrace{(\log(1 - u_{\sigma_{j+1}}^*) - \log(1 - u_{\sigma_j}^*))}_{\geq 0 \text{ since } u_{\sigma_j}^* \text{ is in descending order}}$$

$$\geq \underbrace{(\Pr(C_{\sigma_{1:i^*-1}}|X = \boldsymbol{x}) - \Pr(C_{\sigma'_{1:i^*-1}}|X = \boldsymbol{x}))}_{<0 \text{ according to Condition } 1} \underbrace{(\log u_{\sigma_{i^*}}^* - \log u_{\sigma_{i^*-1}}^*)}_{<0}$$

$$+ \underbrace{(\Pr(C_{\sigma'_{1:i^*-1}}|X = \boldsymbol{x}) - \Pr(C_{\sigma_{1:i^*-1}}|X = \boldsymbol{x}))}_{>0 \text{ according to Condition } 1} \underbrace{(\log(1 - u_{\sigma_{i^*}}^*) - \log(1 - u_{\sigma_{i^*-1}}^*))}_{>0}$$

$$> 0,$$

  which means $\boldsymbol{u}'$ decreases the risk,and thus concludes the proof of this step.

- **Step 2:** We prove that $\{j^*\} = \text{Argmax}_{j \in [J]} u_j^*$, and $u_{j^*}^* = \text{Acc}_{j^*}(\boldsymbol{x})$. Notice that $j^* \in \text{Argmax}_{j \in [J]} u_j^*$:

- Suppose $u_{j^*} = \max_j u_j < \text{Acc}_{j^*}(\boldsymbol{x})$, then we can increase it to $u'_{j^*} = \eta_{j^*}$ with $u'_j = u^*_j$ for $j \neq j^*$, which strictly decreases the risk value due to the properness of log loss.

- Suppose $u_{j^*} = \max_j u_j > \text{Acc}_{j^*}(\boldsymbol{x})$, construct the following $\boldsymbol{u}'$:

$$
u'_j = \begin{cases} \text{Acc}_{j^*}(\boldsymbol{x}), & j = j^*, \\ \max\left(\max_{j \in \{2, \cdots, J\}} \Pr(M_{\sigma_{1:j-1}} \neq Y, M_j = Y | X = \boldsymbol{x}), u^*_j\right), & \text{else.} \end{cases}
$$

Then we can learn the permutation of $\boldsymbol{u}^*$ and $\boldsymbol{u}'$ remain unchanged, and the risk decreases strictly according to the definition of log loss. Furthermore, $\text{Acc}_{j^*}(\boldsymbol{x}) > \text{Acc}_j \geq \Pr(M_{\sigma_{1:j-1}} \neq Y, M_j = Y | X = \boldsymbol{x})$, which means $u^*_j > u^*_{\sigma_2}$.

According to the discussions above, we proved the claim of this step.

Then we can conclude the proof since sigmoid function is invertible. $\qquad\square$

# D    Details of Experiments

We implemented all the methods by Pytorch (Paszke et al., 2019) and conducted all the experiments on NVIDIA H200 GPUs.

## D.1    Experimental Setup on Synthetic Expert Datasets

We use the original training sets of CIFAR-100 and ImageNet, containing 50,000 and 1,281,167 samples, respectively. For each dataset, we split the training set into training and validation subsets with a 9:1 ratio.

**Optimizers**    All models are trained using SGD with a momentum of $0.9$ and a weight decay of $5 \times 10^{-4}$. The initial learning rate is set to $0.1$ and is scheduled by the cosine annealing. Each model is trained for 200 epochs (batch size 128) on CIFAR-100 and 90 epochs (batch size 512) on ImageNet.

**CIFAR-100: Animal Experts**    We consider a subset of CIFAR-100 consisting of 50 **biological classes**, e.g., mammals, fish, and insects. Following common practice, we refer to this subset as the biological (or "animal") subset for simplicity. And we then simulate experts with varying levels of competence for CIFAR-100 in the following ways:

1) **Animal Expert**: Each expert's domain includes 2 disjoint biological classes; accuracy is $94\%$ on domain, $75\%$ on other biological species, and random on the rest.

2) **Overlapped Animal Expert**: We fix the overlap length to 5 and re-define overlapping expert domains for each expert count so that their union exactly covers all biological classes; accuracy follows the same setting as above.

3) **Varying-Accuracy Animal Expert**: Each expert's domain includes 2 disjoint biological classes, with in-domain accuracy linearly increasing from $88\%$ to $94\%$; accuracy on other animals is $75\%$, and random otherwise.

**ImageNet: Dog Experts**    We consider a subset of ImageNet consisting of 120 **dog classes**, e.g., Labrador retriever and German shepherd. Based on this subset, we simulate domain-specialized dog experts with varying levels of competence for ImageNet in the following three ways:

1) **Dog Expert**: Each expert's domain includes 5 disjoint dog classes; accuracy is $88\%$ on domain, $75\%$ on other dog species, and random on the rest.

2) **Overlapped Dog Expert**: We fix the overlap length to 20 and re-define overlapping expert domains for each expert count so that their union exactly covers all dog classes; accuracy follows the same setting as above.

3) **Varying-Accuracy Dog Expert**: Each expert's domain includes 5 disjoint dog classes, with in-domain accuracy linearly increasing from $75\%$ to $88\%$; accuracy on other animals is $75\%$, and random otherwise.

### D.2 Experimental Setup on Real-world Expert Datasets

**Datasets** We further conduct experiments on two datasets with real-world experts, MiceBone (Schmarje et al., 2022) and Chaoyang (Zhu et al., 2022), to demonstrate the practical effectiveness of our PiCCE method.

- The MiceBone dataset consists of 7,240 second-harmonic generation microscopy images categorized into three classes: similar collagen fiber orientation, dissimilar collagen fiber orientation, and not of interest. The dataset contains annotations from 79 professional human annotators, with each image annotated by up to five professional annotators. Among these human annotators, only eight provide labels for the entire dataset. Following Zhang et al. (2026), we consider these eight annotators as human experts. Details of their prediction performance are provided in Table 3. In our experiments, we vary the number of experts, choosing #Exp $\in \{2, 4, 6, 8\}$, and consider the first 2, 4, 6, and 8 experts according to the order in Table 3. The dataset is partitioned into five folds. We use the first four folds for training and the remaining fold for testing, resulting in 5,697 training images and 1,543 test images.

*Table 3.* Prediction accuracy (Acc, %) of the eight selected experts on the MiceBone dataset.

| Expert ID | 047 | 290 | 533 | 534 | 580 | 581 | 966 | 745 |
|---|---|---|---|---|---|---|---|---|
| Train Acc | 84.64 | 85.01 | 87.43 | 88.13 | 81.73 | 85.96 | 87.05 | 85.45 |
| Test Acc | 84.64 | 84.71 | 86.33 | 85.68 | 79.59 | 84.64 | 87.88 | 84.90 |

- The Chaoyang dataset consists of colon histopathology image patches collected at Chaoyang Hospital in China, categorized into four classes: normal, serrated, adenocarcinoma, and adenoma. Each image is annotated by three pathologists with prediction accuracies 87%, 91%, and 99%, respectively. Following Zhang et al. (2023), we use the first two pathologists as experts and exclude the near-oracle annotator. To evaluate settings with different expert cardinalities, we simulate additional experts. Specifically, for #Exp $\in \{4, 6, 8\}$, additional experts are introduced with identical settings to the available ones. Experts' predictions are generated independently.

**Models and Optimizers** We use ResNet-18 as base model for both MiceBone and Chaoyang datasets. All models are trained using AdamW optimizer with an initial learning rate of $3 \times 10^{-4}$ and a weight decay of $5 \times 10^{-4}$. the initial learning rate. Each model is trained for 100 epochs with a batch size of 128.

### D.3 Results with Varying-accuracy Animal Experts

**System Error and Coverage** From Table 4, it can be seen that PiCCE method consistently outperforms the multi-expert CE and OvA surrogate losses under the varying-accuracy synthetic expert setting across both CIFAR-100 and ImageNet datasets, achieving lower system error together with substantially higher coverage. As the number of experts increases, the vanilla framework exhibits pronounced performance degradation, whereas PiCCE remains robust. Overall, these results show that PiCCE effectively improves the overall system performance in multi-expert L2D.

**Classifier Error** As shown in Figure 3, for both the varying-accuracy animal expert setting on CIFAR-100 and the varying-accuracy dog expert setting on ImageNet, the classifier accuracy of CE and OvA degrades steadily as the number of experts increases. In particular, OvA exhibits a pronounced accuracy drop with more experts, while CE also suffers from a consistent decline. In contrast, the PiCCE-based variants maintain stable classifier accuracy across different numbers of experts and consistently outperform their vanilla counterparts on both datasets. This observation further suggests that PiCCE effectively resolves the underfitting induced by expert aggregation in practical multi-expert L2D.

### D.4 Additional Experimental Results

**Violation of Condition 1** Condition 1 assumes that the optimal expert is unique. We further study the robustness of PiCCE when this condition is violated, especially when there exist multiple highly homogeneous or even identical experts. To this end, we conduct additional experiments on CIFAR-100. We begin with the same 4 Animal Experts setting, denoted by $M_{1:4}$, as in the first row of Table 1. We then repeatedly duplicate the original 4 experts, assigning each duplicate the same prediction distribution as its source expert. This expands the expert pool from 4 to 8, 12, 16, and 20 experts. Under this construction, each expert has identical copies, which creates inputs for which multiple experts are equally optimal and

*Table 4.* The mean of the system error (Err, rescaled to 0-100) and coverage (Cov) for 3 trails on the CIFAR-100 and ImageNet datasets. The best performance is highlighted in boldface.

| Expert Pattern | | | Varying-Acc Animal Expert | | | |
| --- | --- | --- | --- | --- | --- | --- |
| Loss Formulation | | | Vanilla | | PiCCE | |
| Dataset | Method | #Exp | Err | Cov | Err | Cov |
| CIFAR-100 | CE | 4 | 18.96 | 75.34 | **18.40** | **77.16** |
| | | 8 | 19.56 | 76.59 | **18.38** | **77.76** |
| | | 12 | 20.18 | 75.31 | **18.21** | **78.09** |
| | | 16 | 22.46 | 76.10 | **18.13** | **78.65** |
| | | 20 | 22.56 | 76.13 | **18.09** | **78.45** |
| | OvA | 4 | 19.07 | 85.04 | **19.01** | **86.33** |
| | | 8 | 20.65 | 84.27 | **18.95** | **85.10** |
| | | 12 | 22.65 | 83.80 | **18.88** | **85.99** |
| | | 16 | 24.77 | 79.21 | **18.83** | **85.85** |
| | | 20 | 26.08 | 78.02 | **18.69** | **85.78** |

| Expert Pattern | | | Varying-Acc Dog Expert | | | |
| --- | --- | --- | --- | --- | --- | --- |
| Loss Formulation | | | Vanilla | | PiCCE | |
| Dataset | Method | #Exp | Err | Cov | Err | Cov |
| ImageNet | CE | 4 | 42.58 | 89.21 | **41.87** | **90.02** |
| | | 8 | 44.27 | 88.82 | **41.28** | **89.93** |
| | | 12 | 46.28 | 88.27 | **41.18** | **90.01** |
| | | 16 | 47.02 | 88.71 | **41.21** | **90.07** |
| | | 20 | 47.98 | 87.81 | **41.39** | **90.10** |
| | OvA | 4 | 45.03 | 88.24 | **42.12** | **89.40** |
| | | 8 | 52.38 | 86.71 | **42.35** | **89.51** |
| | | 12 | 58.13 | 85.16 | **42.51** | **89.26** |
| | | 16 | 61.02 | 84.54 | **42.14** | **89.32** |
| | | 20 | 65.27 | 83.81 | **42.13** | **89.24** |

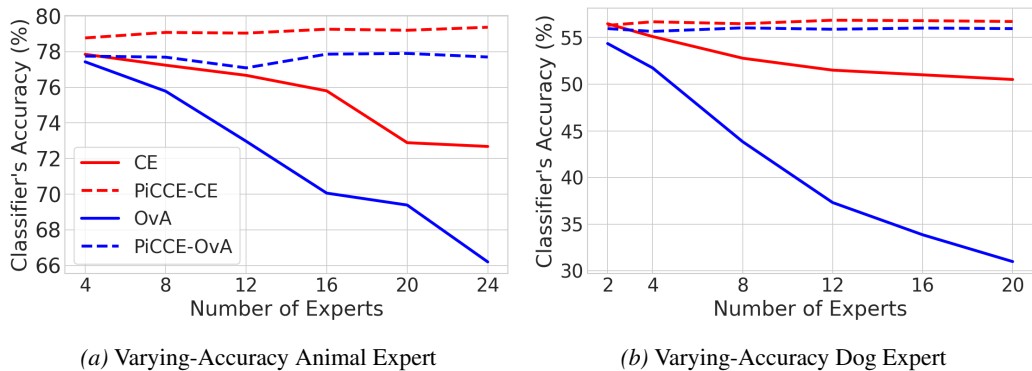

*(a)* Varying-Accuracy Animal Expert          *(b)* Varying-Accuracy Dog Expert

*Figure 3.* We report the classifier's accuracy on CIFAR-100 and ImageNet datasets. Solid lines are the methods from existing formulation (4) while dashed lines are the methods derived from our PiCCE formulation (9).

*Table 5.* The mean of the system error (Err, rescaled to 0-100) and coverage (Cov) for 3 trails on the CIFAR-100 with multiple identical animal experts. The best performance is highlighted in boldface.

| Method | CE | | PiCCE-CE | | OvA | | PiCCE-OvA | |
|---|---|---|---|---|---|---|---|---|
| #Exp | Err | Cov | Err | Cov | Err | Cov | Err | Cov |
| 4 (1 optimal expert) | 18.48 | 74.58 | **18.32** | **77.50** | 19.46 | 83.72 | **19.10** | **86.43** |
| 8 (2 optimal experts) | 19.02 | 74.11 | **18.34** | **77.77** | 20.52 | 83.48 | **19.06** | **86.34** |
| 12 (3 optimal experts) | 19.30 | 73.90 | **18.27** | **77.82** | 21.31 | 82.63 | **19.12** | **86.50** |
| 16 (4 optimal experts) | 20.27 | 73.47 | **18.29** | **77.63** | 21.53 | 81.97 | **19.14** | **86.38** |
| 20 (5 optimal experts) | 21.06 | 72.95 | **18.31** | **77.59** | 21.92 | 81.16 | **19.07** | **86.41** |

*Table 6.* The mean of the system error (Err, rescaled to 0-100) and coverage (Cov) for 3 trails on the CIFAR-100 under Animal Experts setting (#Exp=12) with increasing training data sizes. The training and validation sets are split with a 9:1 ratio. The best performance is highlighted in boldface.

| Method | CE | | PiCCE-CE | |
|---|---|---|---|---|
| Training Sample Size | Err | Cov | Err | Cov |
| 10000 | 41.88 | 56.55 | **32.33** | **59.79** |
| 20000 | 28.17 | 61.12 | **25.12** | **68.23** |
| 30000 | 24.33 | 69.16 | **21.89** | **72.29** |
| 40000 | 21.88 | 72.72 | **18.82** | **75.56** |
| 50000 | 19.08 | 75.18 | **18.19** | **77.43** |

distributionally identical. As shown in Table 5, our proposed PiCCE method remains robust even as the number of identical optimal experts increases. Across all numbers of duplicated experts, PiCCE consistently achieves lower system error and higher coverage than the corresponding baselines, which continue to suffer from underfitting. These results suggest that the uniqueness requirement on the optimal expert may not be essential in practice, and investigating whether Condition 1 can be further relaxed would be an interesting direction for future work.

**Varying Training Sample Size** We empirically compare the performance of CE and PiCCE-CE methods under different training sample sizes on CIFAR-100 under Animal Expert setting (#Exp=12). Specifically, we vary the number of training samples from 10,000 to 50,000, while keeping the training/validation split ratio fixed at 9:1. The results in Table 6 indicate that both methods improve as the training sample size increases, while PiCCE-CE maintains a clear advantage over CE by achieving both lower system error and higher coverage. The improvement is particularly large when the training set is small, suggesting that PiCCE is more robust when training data is limited.

# E   Finite-sample Guarantee

## E.1   Connections between $\widetilde{\ell}_\phi$ and $\phi$

While a common practice of finite-sample analysis for loss minimization depends on the assumption on the regularity of multiclass loss $\phi$, e.g., boundedness/Lipschitzness, it is still unclear whether $\widetilde{\ell}_\phi$ will inherit these properties. The following conclusion confirmed that $\widetilde{\ell}_\phi$ will preserve the regularity of $\phi$.

**Lemma E.1.** *Suppose $\phi$ satisfies the continuity and symmetry condition in Theorem 4.2. If the multiclass loss $\phi$ is non-negative and bounded by $B$, then $\widetilde{\ell}_\phi$ is non-negative and bounded by $2B$. If $\phi$ is $L_\phi$-Lipschitz continuous w.r.t. $\boldsymbol{\theta}$ and , then $\widetilde{\ell}_\phi$ is $2L_\phi$-Lipschitz continuous.*

*Proof.* Recall the definition of PiCCE:

$$\widetilde{\ell}_\phi(\boldsymbol{\theta}, y, \boldsymbol{m}) = \phi(\boldsymbol{\theta}, y) + \phi\Big(\boldsymbol{\theta}, \operatorname*{argmax}_{j \in [\boldsymbol{m}=y]} \theta_{j+K} + K\Big).$$

Since $0 \le \phi(\boldsymbol{\theta}, y) \le B$, we can conclude $0 \le \widetilde{\ell}_\phi(\boldsymbol{\theta}, y, \boldsymbol{m}) \le 2B$.

Then we focus on the Lipschitzness of $\widetilde{\ell}_\phi$. For any $\boldsymbol{\theta}, \boldsymbol{\theta}' \in \mathbb{R}^{K+J}$, for a fixed $i \in \mathrm{argmax}_{j \in [\boldsymbol{m}=y]} \theta_{j+K}$, we have:

$$\widetilde{\ell}_\phi(\boldsymbol{\theta}, y, \boldsymbol{m}) = \phi(\boldsymbol{\theta}, y) + \phi(\boldsymbol{\theta}, i + K).$$

For a fixed $i' \in \mathrm{argmax}_{j \in [\boldsymbol{m}=y]} \theta'_{j+K}$, denote by $\boldsymbol{\theta}''$ a new vector generated by swapping $\theta'_{i+K}$ and $\theta'_{i'+K}$. According to the symmetry, we have:

$$\begin{aligned}
\widetilde{\ell}_\phi(\boldsymbol{\theta}', y, \boldsymbol{m}) &= \phi(\boldsymbol{\theta}', y) + \phi(\boldsymbol{\theta}', i' + K) \\
&= \phi(\boldsymbol{\theta}'', y) + \phi(\boldsymbol{\theta}'', i + K) \\
&= \phi(\boldsymbol{\theta}', y) + \phi(\boldsymbol{\theta}'', i + K)
\end{aligned}$$

Denote $a = \theta_{K+i}, b = \theta_{K+i'}, a' = \theta'_{K+i}$, and $b' = \theta'_{K+i'}$. Since $i \in \mathrm{argmax}_{j \in [\boldsymbol{m}=y]} \theta_{K+j}$ and $i' \in \mathrm{argmax}_{j \in [\boldsymbol{m}=y]} \theta'_{K+j}$, we have $a \geq b$ and $b' \geq a'$. Since $\boldsymbol{\theta}''$ is obtained from $\boldsymbol{\theta}'$ by swapping the coordinates $K + i$ and $K + i'$, all coordinates except these two are unchanged. Hence

$$\|\boldsymbol{\theta} - \boldsymbol{\theta}''\|_1 - \|\boldsymbol{\theta} - \boldsymbol{\theta}'\|_1 = |a - b'| + |b - a'| - |a - a'| - |b - b'|.$$

For any $a \geq b$ and $c \geq d$, we have $|a - c| + |b - d| \leq |a - d| + |b - c|$. Taking $c = b'$ and $d = a'$, and we obtain

$$|a - b'| + |b - a'| \leq |a - a'| + |b - b'|.$$

Therefore, $\|\boldsymbol{\theta} - \boldsymbol{\theta}''\|_1 \leq \|\boldsymbol{\theta} - \boldsymbol{\theta}'\|_1$.

By the symmetry of $\phi$ with respect to the last $J$ coordinates, we have $\phi(\boldsymbol{\theta}', y) = \phi(\boldsymbol{\theta}'', y)$ and $\phi(\boldsymbol{\theta}', K+i') = \phi(\boldsymbol{\theta}'', K+i)$. Therefore,

$$\begin{aligned}
&|\widetilde{\ell}_\phi(\boldsymbol{\theta}, y, \boldsymbol{m}) - \widetilde{\ell}_\phi(\boldsymbol{\theta}', y, \boldsymbol{m})| \\
&= |\phi(\boldsymbol{\theta}, y) + \phi(\boldsymbol{\theta}, K + i) - \phi(\boldsymbol{\theta}'', y) - \phi(\boldsymbol{\theta}'', K + i)| \\
&\leq |\phi(\boldsymbol{\theta}, y) - \phi(\boldsymbol{\theta}'', y)| + |\phi(\boldsymbol{\theta}, K + i) - \phi(\boldsymbol{\theta}'', K + i)| \\
&\leq 2L\|\boldsymbol{\theta} - \boldsymbol{\theta}''\|_1 \leq 2L\|\boldsymbol{\theta} - \boldsymbol{\theta}'\|_1.
\end{aligned}$$

Thus, $\widetilde{\ell}_\phi(\cdot, y, \boldsymbol{m})$ is $2L$-Lipschitz. $\qquad\square$

## E.2 Proof

Let $\mathcal{G} :\subseteq \mathcal{X} \to \mathbb{R}^{K+J}$ be the model class and each of its dimension is $\mathcal{G}_y \subseteq \mathcal{X} \to \mathbb{R}$. Assume there exists $C_{\boldsymbol{g}} > 0$ that $\sup_{\boldsymbol{g} \in \mathcal{G}} \|\boldsymbol{g}\|_\infty \leq C_{\boldsymbol{g}}$ and $C_\phi > 0$ that $\sup_{\|\boldsymbol{\theta}\|_\infty \leq C_{\boldsymbol{g}}} \phi(\boldsymbol{\theta}, y) \leq C_\phi$ for any $y$. We also assume $\phi(\boldsymbol{\theta}, y)$ is $L_\phi$-Lipschitz *w.r.t.* $\boldsymbol{\theta}$. Given $n$ samples $\{z_i\}_{i=1}^n$ that each $z_i := (\boldsymbol{x}_i, y_i, \boldsymbol{m}_i)$ is drawn from $\mathcal{D}$ independently, and we slightly abuse the notation by the following simplification:

$$\widetilde{\ell}_\phi(\boldsymbol{g}; z_i) := \widetilde{\ell}_\phi(\boldsymbol{g}(\boldsymbol{x}_i), y_i, \boldsymbol{m}_i).$$

Further denote by $\hat{\boldsymbol{g}}$ the empirical minimizer of the following empirical risk:

$$\widehat{R}_{\widetilde{\ell}_\phi}(\boldsymbol{g}) = \frac{1}{n} \sum_{i=1}^n \widetilde{\ell}_\phi(\boldsymbol{g}; z_i). \tag{15}$$

Then we have the following guarantee:

**Theorem E.2.** *For any $\delta > 0$, the following inequality holds with probability at least $1 - \delta$:*

$$R_{\widetilde{\ell}_\phi}(\hat{\boldsymbol{g}}) - \inf_{\boldsymbol{g} \in \mathcal{G}} R_{\widetilde{\ell}_\phi}(\boldsymbol{g}) \leq 8\sqrt{2} L_\phi \sum_{y=1}^{K+J} \mathfrak{R}_n(\mathcal{G}_y) + 8C_\phi \sqrt{\frac{\ln \frac{2}{\delta}}{2n}},$$

*where $\mathfrak{R}_n(\mathcal{G}_y)$ is the Rademacher complexity of $\mathcal{G}_y$.*

*Proof.* Firstly, we define the following class:

$$\mathcal{L} = \left\{ z \to \widetilde{\ell}_\phi(\boldsymbol{g}; z) : \boldsymbol{g} \in \mathcal{G} \right\}.$$

Notice that $0 \leq \widetilde{\ell}_\phi(\boldsymbol{g}; z) \leq 2C_\phi$ for all $\boldsymbol{g} \in \mathcal{G}$ and $z$ according to Lemma E.1, we can conclude from the standard symmetrization and McDiarmid's inequality (McDiarmid et al., 1989) that with probability at least $1 - \delta$,

$$\sup_{\boldsymbol{g} \in \mathcal{G}} \left| R_{\widetilde{\ell}_\phi}(\boldsymbol{g}) - \widehat{R}_{\widetilde{\ell}_\phi}(\boldsymbol{g}) \right| \leq 2\mathfrak{R}_n(\mathcal{L}) + 4C_\phi \sqrt{\frac{\ln \frac{2}{\delta}}{2n}},$$

According to the vector-valued Talagrand's inequality (Maurer, 2016, Corollary 1) and Lemma E.1, we can reach the following conclusion:

$$\mathfrak{R}_n(\mathcal{L}) \leq 2\sqrt{2} L_\phi \sum_{y=1}^{K+J} \mathfrak{R}_n(\mathcal{G}_y).$$

Then we bound the uniform convergence via the definition of empirical risk minimizer:

$$R_{\widetilde{\ell}_\phi}(\hat{\boldsymbol{g}}) - R_{\widetilde{\ell}_\phi}(\boldsymbol{g}) = R_{\widetilde{\ell}_\phi}(\hat{\boldsymbol{g}}) - \widehat{R}_{\widetilde{\ell}_\phi}(\hat{\boldsymbol{g}}) + \widehat{R}_{\widetilde{\ell}_\phi}(\hat{\boldsymbol{g}}) - \widehat{R}_{\widetilde{\ell}_\phi}(\boldsymbol{g}) + \widehat{R}_{\widetilde{\ell}_\phi}(\boldsymbol{g}) - R_{\widetilde{\ell}_\phi}(\boldsymbol{g}) \leq 2 \sup_{\boldsymbol{g}' \in \mathcal{G}} \left| R_{\widetilde{\ell}_\phi}(\boldsymbol{g}') - \widehat{R}_{\widetilde{\ell}_\phi}(\boldsymbol{g}') \right|.$$

Taking the infimum over $\boldsymbol{g} \in \mathcal{G}$ gives

$$R_{\widetilde{\ell}_\phi}(\hat{\boldsymbol{g}}) - \inf_{\boldsymbol{g} \in \mathcal{G}} R_{\widetilde{\ell}_\phi}(\boldsymbol{g}) \leq 2 \sup_{\boldsymbol{g}' \in \mathcal{G}} \left| R_{\widetilde{\ell}_\phi}(\boldsymbol{g}') - \widehat{R}_{\widetilde{\ell}_\phi}(\boldsymbol{g}') \right|.$$

Combining this inequality with the conclusions above and we can learn:

$$R_{\widetilde{\ell}_\phi}(\hat{\boldsymbol{g}}) - \inf_{\boldsymbol{g} \in \mathcal{G}} R_{\widetilde{\ell}_\phi}(\boldsymbol{g}) \leq 4\mathfrak{R}_n(\mathcal{L}) + 4C_{\widetilde{\ell}_\phi} \sqrt{\frac{\ln \frac{2}{\delta}}{2n}}$$

$$\leq 8\sqrt{2} L_\phi \sum_{y=1}^{K+J} \mathfrak{R}_n(\mathcal{G}_y) + 8C_\phi \sqrt{\frac{\ln \frac{2}{\delta}}{2n}}.$$

$\square$

# F    Limitations and Future Directions

Our current setting is restricted to multi-expert L2D without extra costs. As a result, it remains unclear whether the existence of such costs would cause underfitting. A promising direction for future work is therefore to study this setting and develop compatible solutions. Another important direction is to make use of more existing loss functions. At present, our consistency analysis focuses primarily on cross-entropy-like surrogate losses. However, original multi-expert L2D is naturally compatible with any calibrated multiclass loss, which suggests that a broader class of objectives is worth investigating.

