# OpenReview forum: "When More Experts Hurt: Underfitting in Multi-Expert Learning to Defer"
_ICML.cc/2026/Conference — ICML 2026 regular_

### Official Review · Reviewer_28wb · 2026-03-09

**Soundness:** 3
**Presentation:** 4
**Significance:** 2
**Originality:** 3
**Overall Recommendation:** 4
**Confidence:** 3

**Summary:**

Learning to defer is a framework that allows a classifier to abstain from making a prediction and defer the decision to an expert. Recent work has extended this setting to a multi-expert scenario. However, existing methods in this setting often suffer from poor performance, especially as the number of experts increases, as they tend to lead to underfitting of the classifier. In this paper, the authors theoretically explain the source of this underfitting and propose a new method, PiCCE, to address this issue. The key idea is to identify the best expert and reduce the multi-expert problem to a single-expert problem involving this optimal agent.

**Compliance With Llm Reviewing Policy:**

Affirmed.

**Final Justification:**

The authors have addressed my main concerns and convinced me, particularly regarding the question of finite-sample guarantees and the role of $\theta$.

**Key Questions For Authors:**

In the definition of PiCCE (Definition 1, Equation 9), the loss depends on the model’s internal beliefs through the parameter $\theta$. Could this lead to a self-reinforcement mechanism or confirmation bias, potentially degrading performance if the model is poorly initialized or if the number of training samples is small? In particular, in your experiments on CIFAR-100 and ImageNet, the number of training samples used is not specified. Could you please clarify this?

**Limitations:**

yes

**Strengths And Weaknesses:**

**Strengths:**

1. The paper studies an interesting and relatively new problem: multi-expert learning to defer, which is a promising research direction and more challenging than the single-expert setting.

2. The authors explain (novelly) why existing methods for multi-expert learning to defer underperform, showing that they tend to lead to underfitting of the classifier.

3. The proposed solution is simple and intuitive: identify the best expert and reduce the problem to a single-expert learning-to-defer setting with this optimal expert.

4. The experiments are well conducted, and the plots are clear and informative.

**Weaknesses:**

1. The theoretical setting is somewhat limited, as the results are stated in probability and do not provide finite-sample guarantees. Since, in practice, we only have access to a finite amount of data, it may be difficult (if not impossible) to properly handle small probabilities in such settings.

2. Although the results improve upon existing methods (which tend to lead to underfitting and thus decrease performance), I still find them somewhat unconvincing. In particular, the performance does not increase with the number of experts, since the problem is ultimately reduced to a single-expert-like setting. Intuitively, adding more experts should allow for improved performance. Instead of selecting a single expert, it might have been more beneficial to leverage the collaborative potential of multiple experts. Have the authors considered deferring to a subset of experts, rather than just one?

---

> ### Author Rebuttal · Authors · 2026-03-31
>
> We sincerely thank the reviewer for the detailed review! We address the main concerns below.
>
> **Q1. Finite-sample guarantees.**
>
> **A1.** Thank you for raising this concern! Since PiCCE follows empirical risk minimization, we can show it converges to the optimal solution in $O_{p}(1/\sqrt{n})$ and $n$ is the sample size.
>
> Due to the length limit, the full proof will be included in the revised appendix.
>
> Proof sketch: under conditions of Thm 4.2, PiCCE inherits standard properties of $\phi$ (e.g., boundedness/Lipschitzness).
> We then apply McDiarmid's inequality and Talagrand's contraction lemma to bound the excess risk via the Rademacher complexity of the scoring function class, which decays as $O(1/\sqrt{n})$.
>
> **Q2-(a). The performance does not increase with the number of experts.**
> **-----(b). Have the authors considered deferring to a subset of experts?**
>
> **A2-(a).** Thank you for the constructive questions! We would like to first clarify the optimal solution under the standard L2D formulation. As shown in the target system loss Eq. (1) and its Bayes optimal solution Eq. (3), the optimal system selects the single best entity (classifier or expert) for each input. Therefore, the optimal performance is determined by the most accurate expert in the pool rather than the collective set. This is not guaranteed to improve in general, as newly added experts may not necessarily outperform the existing ones.
>
> However, it is a natural scenario that enlarging the expert pool introduces stronger experts. We simulate this on CIFAR100, where each expert’s domain includes 2 disjoint biological classes.
> The first 4 experts have domain/other-biology/remaining accuracy of 80/70/1% and each subsequent group of 4 newly added experts has 5% higher accuracy in all classes than previously added ones. Averaged results over 3 trials:
>
> |Method| CE||PiCCE-CE||
> |:-:|:-:|:-:|:-:|:-:|
> |#Exp|Err|Cov|Err|Cov|
> |4|18.71|77.37|**17.96**|**78.69**|
> |8|19.56|74.95|**17.33**|**76.91**|
> |12|19.75|74.04|**16.70**|**75.21**|
> |16|20.42|71.43|**15.37**|**73.90**|
>
> It shows that PiCCE successfully leverages the benefit of a stronger expert pool, with system error decreasing with more capable experts. In contrast, vanilla CE, despite equally utilizing all expert predictions, fails to benefit from better experts due to underfitting. It demonstrates that PiCCE can effectively leverage the benefit of a growing expert pool.
>
> **-----(b).** This is an interesting perspective. We note that the standard L2D formulation focuses on end-to-end prediction systems (Mozannar & Sontag, 2020; Verma & Nalisnick, 2022; Verma et al., 2023; Mao et al., 2024a; Mao et al., 2025) that must select exactly one entity per input to produce a final class-label decision. We appreciate this suggestion and will discuss the potential of leveraging multiple experts for tasks beyond standard L2D.
>
> **Q3. Could the dependence on θ degrade performance under poor initialization or limited training data?**
>
> **A3.** Thanks for the insightful question! We believe the dependence on θ does not significantly degrade performance under these scenarios, as supported by both theory and experiments.
>  As demonstrated by the bound discussed in Q1 and consistency analysis in Sec. 5, while our method relies on θ, it provably converges to the optimal solution Eq. (3) as sample size increases.
> We empirically evaluate the performance on CIFAR100+Animal Experts (#Exp=12) with increasing training data. Averaged results over 3 trials:
>
> |Method| CE||PiCCE-CE||
> |:-:|:-:|:-:|:-:|:-:|
> |#Data (train:val=9:1)|Err|Cov|Err|Cov|Err|Cov|Err|Cov|
> |10000|41.88|56.55|**32.33**|**59.79**|
> |20000|28.17|61.12|**25.12**|**68.23**|
> |30000|24.33|69.16|**21.89**|**72.29**|
> |40000|21.88|72.72|**18.82**|**75.56**|
> |50000|19.08|75.18|**18.19**|**77.43**|
>
> It shows that PiCCE improves steadily with increasing data size and consistently outperforms the vanilla approach (which does not depend on θ) across all sizes, which supports the dependence on θ in PiCCE for mitigating underfitting.
>
> To further investigate whether this dependence is sensitive to initializations, we run experiments on CIFAR100 (#Data=50000)+Animal Experts (#Exp=12) with various initializations, in addition to our default Kaiming normal.
> Averaged results over 3 trials show that PiCCE remains robust and consistently outperforms CE across all initializations.
>
> |Method|CE||PiCCE-CE||
> |:-:|:-:|:-:|:-:|:-:|
> |Initialization|Err|Cov|Err|Cov|Err|Cov|Err|Cov|
> |Kaiming normal|19.08|75.18|**18.19**|**77.43**|
> |Xavier uniform|19.17|75.86|**17.85**|**78.45**|
> |Kaiming uniform|19.27|75.68|**17.71**|**78.49**|
> |Gaussian normal| 19.00|73.35|**17.82**|**78.09**|
>
>  **Q4. The authors should specify number of training samples used in the experiment.**
>
>  **A4.** Thank you very much for pointing this out!
>  We use original training sets of CIFAR100 (50000) and ImageNet (1281167), each split into training/validation with a 9:1 ratio. We will clarify this in the revised version.

---

> > ### Author Rebuttal · Reviewer_28wb · 2026-04-02
> >
> > I thank the authors for their response, which helped clarify several key aspects of the paper and address some of my initial concerns. I will raise my score accordingly.

---

### Official Review · Reviewer_V6aq · 2026-03-10

**Soundness:** 3
**Presentation:** 3
**Significance:** 2
**Originality:** 3
**Overall Recommendation:** 4
**Confidence:** 3

**Summary:**

This paper studies the multi-expert learning-to-defer problem, asking why it intrinsically suffers from classifier underfitting as the expert pool grows, and how to effectively overcome this fundamental flaw. The authors identify that the standard multi-expert surrogate risk contains an O(J) expert aggregation term that severely flattens the label distribution, making the correct class harder to identify. The paper proposes PiCCE, a surrogate-based method that dynamically prunes the candidate expert set to only empirically correct experts utilizing ground-truth evaluations. The theoretical results shows that the continuous PiCCE loss can compresses the expert accuracy summation into an O(1) term, and is Bayes-consistent and natively extracts true optimal expert accuracies under some assumptions. Experimentally, detailed assessments on synthetic biological/dog domain subsets, alongside real-world medical datasets robustly confirm these results.

**Compliance With Llm Reviewing Policy:**

Affirmed.

**Final Justification:**

The additional baseline comparisons and robustness experiments fully resolve my concerns. Therefore, I am maintaining my score.

**Key Questions For Authors:**

1. Could you provide empirical comparisons against at least one recent state-of-the-art two-stage or post-hoc multi-expert L2D method？
2. How does the theoretical consistency and empirical stability of the L2D system behave if Condition 1 is violated, particularly in scenarios where the optimal expert is not unique (e.g., the pool contains two highly homogeneous or perfectly identical experts)?

**Limitations:**

yes

**Strengths And Weaknesses:**

Strengths:
1.	The progression of the paper is exceptionally natural and compelling. It smoothly transitions from showing why naive intermediate estimators fail (due to discontinuity) to proposing a highly targeted and conceptually sound solution.
2.	The algorithm PiCCE is simple and direct.
3.	The theoretical foundations are rigorous and the experimental design is thorough.

Weaknesses:
1.	The experimental evaluation currently focuses strictly on comparing the proposed PiCCE formulation against its vanilla surrogate counterparts (multi-expert CE and OvA). There lacks comparisons to at least one state-of-the-art two-stage method as a baseline.
2.	Theorem 5.2 anchors the consistency of the L2D system on Condition 1. The manuscript would greatly benefit from a more detailed discussion on what happens when this assumption is violated. For example, how does the system behave if there are two highly homogeneous (or perfectly identical) experts with the highest accuracy?

---

> ### Author Rebuttal · Authors · 2026-03-31
>
> We sincerely thank the reviewer for the constructive and positive review! We address the questions and weaknesses below.
>
> **Q1. The state-of-the-art two-stage method should be considered as a baseline.**
>
> **A1.** Thank you for your constructive suggestion! Following this suggestion, we further evaluate two-stage methods: Eq. (3) in (Mao et al., 2023) and Eq. (5) in (Mao et al., 2025) on CIFAR-100 under the Animal Experts setting, where we use $\ell=\ell_{\log}$ in Eq. (3) and $\Phi(u)=-\log(u)$ in Eq. (5) as in the original papers. We use the same model as in our paper and follow other configurations as in (Mao et al., 2023) and (Mao et al., 2025).
> The averaged results over 3 trials are reported below:
>
> |Method| (Mao et al., 2023) ||(Mao et al., 2025)||
> |:-:|:-:|:-:|:-:|:-:|
> |#Exp|System Err|Cov|System Err|Cov|
> |4|19.70|80.43|20.11|90.25|
> |8|19.96|78.02|20.45|88.34|
> |12|20.02|76.76|20.83|87.24|
> |16|20.35|76.16|21.11|85.93|
> |20|20.56|75.47|21.42|85.41|
>
> According to the results above, our proposed PiCCE method consistently outperforms these two-stage baselines across different numbers of experts, further demonstrating its empirical effectiveness.
>
> **Q2. How does the proposed method behave when Condition 1 is violated, especially in the presence of two highly homogeneous or perfectly identical experts?**
>
> **A2.** Thank you for your insightful question! We further conduct experiments on CIFAR-100+Animal Experts to simulate the presence of multiple identical experts.
>
> **Setup:** We begin with the same 4 Animal experts setting $M_{1:4}$ as in the first row of Table 1. To emulate multiple identical experts, we repeatedly duplicate the original 4 experts,
> assigning each duplicate the same prediction distribution as its source expert. This expands the expert pool from 4 to 8, 12, 16, and 20 experts.
> Under this construction, each expert has identical copies, which creates inputs for which multiple experts are equally optimal and distributionally identical.
> We use this controlled setup to evaluate the behavior of our method when Condition 1 is violated. The averaged results over 3 trials are reported below:
>
> |Method|CE||PiCCE-CE||OvA||PiCCE-OvA||
> |:-:|:-:|:-:|:-:|:-:|:-:|:-:|:-:|:-:|
> |#Exp|Err|Cov|Err|Cov|Err|Cov|Err|Cov|
> |4 (1 optimal expert)|18.48|74.58|**18.32**|**77.50**|19.46|83.72|**19.10**|**86.43**|
> |8 (2 optimal experts)|19.02|74.11|**18.34**|**77.77**|20.52|83.48|**19.06**|**86.34**|
> |12 (3 optimal experts)|19.30|73.90|**18.27**|**77.82**|21.31|82.63|**19.12**|**86.50**|
> |16 (4 optimal experts)|20.27|73.47|**18.29**|**77.63**|21.53|81.97|**19.14**|**86.38**|
> |20 (5 optimal experts)|21.06|72.95|**18.31**|**77.59**|21.92|81.16|**19.07**|**86.41**|
>
>
> According to the results above, our proposed PiCCE method remains robust even as the number of identical optimal experts increases,
> while the baselines still suffer from underfitting.
> These results suggest that the uniqueness requirement on the optimal expert may not be essential in practice,
> and investigating whether Condition 1 can be further relaxed would be an interesting direction for future work.

---

> > ### Author Rebuttal · Reviewer_V6aq · 2026-04-02
> >
> > The additional baseline comparisons and robustness experiments fully resolve my concerns. Therefore, I am maintaining my score.

---

### Official Review · Reviewer_B77a · 2026-03-12

**Soundness:** 2
**Presentation:** 2
**Significance:** 3
**Originality:** 3
**Overall Recommendation:** 5
**Confidence:** 4

**Summary:**

The paper proposes a novel approach to mitigate underfitting in multi-expert learning-to-defer baselines. The core idea is to reduce the multi-expert setting to a single-expert scenario to apply mitigation strategies. Empirical evidence suggests the improvements of the proposed approach.

**Compliance With Llm Reviewing Policy:**

Affirmed.

**Final Justification:**

I appreciated reading this work. I think the idea has more merits than shortcomings and is worth acceptance.

**Key Questions For Authors:**

I have a couple of questions for the authors:
1. I would like the authors to discuss the weaknesses I highlighted;
2. I agree with the authors that some surrogate losses (the ones mentioned in Section 2.2) might be prone to underfitting when $c$ increases, still some more recent losses - e.g., the realizable-consistent losses in (Mozannar et al., 2023; Mao et al., 2024b) -  seem slightly different to me. In particular, in these losses, the objective can be written (up to notation) as
$$\ell(y, f_y(x))c(x,y) +(1-c(x,y))\ell(y, f_y(x)+f_\perp(x)),$$
where $\ell$ is a surrogate loss, $f_y(x)$ is the logit associated to the ground truth class and $f_\perp$ is the logit associated to the defer head. In this case, it seems to me that underfitting might be harder to control, as increasing the cost could force the model to place more "weight" on the ground-truth head rather than on the expert's head.  Could the authors clarify whether the underfitting mechanism presented in Sec. 2.2 (e.g., via an induced label-smoothing / margin-shrink effect) applies to these realizable-consistent surrogates as well? If not, a brief discussion delimiting which surrogate families exhibit the $c$-driven underfitting would strengthen Sec. 2.2.

**Limitations:**

There is no limitation section, so I would encourage the authors to acknowledge some of the limits of their current work.

**Strengths And Weaknesses:**

The paper's main strengths are:

1. **Solid theoretical mitigation strategy.** The paper provides a principled method to mitigate underfitting in multi-expert learning to defer baselines. The arguments are mostly correct (I have a few remarks about a couple of points in the proofs). Moreover, I think reducing the multi-expert problem to a single expert one is a clever strategy;
2. **The paper is well-written.** I think the paper is nicely presented overall, providing some intuition of why underfitting might occur;
3. **The paper is potentially impactful.** The multi-expert learning-to-defer setting is overlooked at the moment, but I think this paper can be impactful in the community by shedding light on the under-explored issue of underfitting.


The paper's main weaknesses are:

1.  **Baselines not considered in the experimental evaluation.** I might be missing something, but it seems to me that recent loss functions - introduced in Mao et al., 2025 - are not considered in the experimental evaluation. I think including them could improve the paper further;
2.  **Only synthetic experts. ** While the authors present results fully controlling experts, I think some datasets with multiple expert annotations are available in the L2D literature (see e.g., Mozannar et al., 2023 or Palomba et al., 2025 for some datasets). Using real-world data with real human experts could make results more convincing;
3. **Some small adjustments in the proofs might be required.** I have a couple of small concerns in some passages of the proofs.
    - *Theorem 4.2 (Case 1)*, I might have missed it, but it seems like the norm for defining $\Delta$ is not specified. This likely doesn’t change the result, but making it explicit would clarify which constants are needed in the subsequent bounds. Moreover, I would opt for considering $\frac{\Delta}{2}$ so to retrieve $\varepsilon$ rather than $2\varepsilon$.
    - *Theorem 4.2 (Case 2)*, my understanding is that symmetry is used together with the fact that maximizers are tied at $\theta_0$: swapping tied expert logits leaves $\theta_0$ unchanged, hence the corresponding loss components coincide.  If this is the case, it may help to phrase this explicitly (rather than ``Because ``$\phi$`` is symmetric w.r.t. its last J inputs``), and add one intermediate line showing the permutation/swap argument.
    - *Lemma 5.1* At line 789, the authors claim that ``The last inequality is obtained since log loss is a proper loss.`` I think the result should hold, but I would try to add a couple of steps to clarify better that inequality.

---

> ### Author Rebuttal · Authors · 2026-03-31
>
> We sincerely thank the reviewer for the thorough and positive review! The questions and corresponding answers are summarized below:
>
> **Q1. The authors should consider more recent loss functions in the experimental evaluation.**
>
> **A1.** Many thanks for your constructive suggestion! Following this comment, we further evaluate the $L_{\mathrm{MAE}}$ loss from (Mao et al. 2025) on the CIFAR-100 Animal Experts setting, which is also the loss explicitly evaluated in (Mao et al. 2025).
> We use the same configuration as in our experiments, except for the optimizer: we adopt Adam with a learning rate of 1e-3 as in (Mao et al. 2025), since it performs better than SGD for this loss.
> The averaged results over 3 trials are reported below.
>
> | #Exp | System Err  | Cov |
> |:--:|:--:|:--:|
> | 4 | 26.01|  59.45 |
> | 8 | 26.96|  60.34 |
> | 12| 28.03|  58.27 |
> | 16| 27.84|  58.89 |
> | 20| 28.35|  58.52 |
>
> According to the results above, we can see that our proposed PiCCE with CE or OvA losses outperforms $L_{MAE}$, likely because it
> is primarily designed to achieve both Bayes- and H-consistency, while its uniformly bounded nature may make optimization in deep models less favorable due to comparatively weaker gradient signals.
>
> **Q2. Real-world human expert annotations should be considered in the experiment.**
>
> **A2.** Thank you very much for this helpful suggestion! We would like to clarify that our experiments already include real-world expert annotations. In particular, **MiceBone** and **Chaoyang** are multi-class datasets with real-world expert annotations. Currently, we mention this point in line 427 and Appendix D.2, which is not sufficiently highlighted. In the revised version, we will highlight it more clearly in the main text for clearer presentation.
>
>
> **Q3.** **Some small adjustments in the proofs might be required.**
>
> **A3.** Thank you for your insightful comment and for carefully proofreading our proof! In the proof of Theorem 4.2, we will specify in Case 1 that the norm for defining $\Delta$ is the 2-norm and consider $\Delta/2$ in the derivation; in Case 2, we will elaborate on the use of the symmetry of $\phi$.
> We will also emphasize in Lemma 5.1 that the inequality comes from the definition of strictly proper losses, that the loss is minimized iff the estimate equals the class probability.
>
>
> **Q4.**  **Does the c-driven underfitting mechanism in Sec. 2.2 also applies to realizable-consistent surrogate losses?**
>
> **A4.** Thank you very much for raising this interesting concern!
> Existing analyses of $c$-driven underfitting (Narasimhan et al., 2022; Liu et al., 2024) focus on the methods (mentioned in Sec. 2.2) that belong to the framework in (Charusaie et al., 2022), which casts the single-expert L2D problem as K+1 class classification.
> The class of realizable-consistent surrogates (Mozannar et al., 2023; Mao et al., 2024b) is not constructed according to this framework and thus is not covered by the $c$-driven underfitting mechanism, and studying their behaviour remains an interesting future direction of single-expert L2D.
> We will emphasize this focus in Sec. 2.2 to include the relevant literature on realizable-consistent surrogate losses (Mozannar et al., 2023; Mao et al., 2024b).
>
> *Mozannar, Hussein, et al. "Who should predict? exact algorithms for learning to defer to humans." AISTATS, 2023.*
>
> *Mao, Anqi, et al. "Realizable $H$-consistent and Bayes-consistent loss functions for learning to defer." NeurIPS, 2024.*
>
> **Q5.**  **The authors are encouraged to add a limitation section.**
>
> **A5.** Many thanks for your helpful suggestion! We will add a limitation section to the revised manuscript.

---

> > ### Author Rebuttal · Reviewer_B77a · 2026-03-31
> >
> > I thank the authors for their reply. My concerns are resolved and I am willing to increase my score.

---

### Decision · Program_Chairs · 2026-04-30

**Decision:**

Accept (regular)

**Comment:**

The paper studies a multi-expert learning-to-defer framework. They focus on how to avoid underfitting as the number of experts grows, showing how that this problem comes from expert aggregation. They propose a method to select experts and thus reduce to a single-expert problem. They theoretically and empirically demonstrate the performance of this method.

The reviewers appreciated the clean explanation of why existing approaches fail, the principled nature of the proposed solution, and the clear presentation. The additional experiments in the author rebuttal addressed concerns about loss functions and baselines. Thus, I recommend acceptance. We encourage the authors to incorporate the additional experiments into the final version of the paper.